# Spin switchable optical phenomena in Rashba band structures through intersystem crossing in momentum space in solution-processing 2D-superlattice perovskite film

Bogdan Dryzhakov[1], Yipeng Tang[1], Jong Keum [2,3], Haile Ambaye[2], Jinwoo Kim [4], Tae-Woo Lee [4], Valeria Lauter [2] ✉ & Bin Hu [1] ✉

Spin-switchable phenomena are a critical element for the development of spintronic and chiroptic devices. Herein we combine a 2D-superlattice perovskite (4,4-DFPD$_2$PbI$_4$) film with a ferromagnetic cobalt (Co) layer to form a multiferroic perovskite/Co interface, and demonstrate spin-switchable circularly polarized luminescence (CPL) between right-handed $\sigma^+$ and left-handed $\sigma^-$ polarizations. When the ferromagnetic spins of Co at the Co/perovskite interface are altered between positive and negative magnetic field directions, the CPL from the 2D-superlattice perovskite switches from $\sigma^+$ to $\sigma^-$ polarization. The magnetic field effects present a unique method to confirm that CPL is generated by the circular-orbital momentum of light-emitting excitons within Rashba band structures, eliminating artifacts involving structural birefringence. Our polarized neutron reflectometry measurements confirm a super long-range spin-orbit interaction occurring in the 2D-superlattice perovskite films. The temperature dependence of spin-switchable phenomenon indicates an extraordinarily long orbital polarization lifetime, reaching microseconds at room temperature and milliseconds at 5 K.

Spin-switchable phenomena function as the critical element to demonstrate spintronic, orbitronic, and chiroptic behaviors demanded for advancing energy, sensing, and computing technologies. Two-dimensional (2D) perovskites are multifunctional, solution-processed semiconductors known for their attractive photovoltaic and detection properties[1–4], distinguished with excellent light-emitting[5,6] and transport behaviors[7,8]. Their structure consists of inorganic metal-halide octahedral layers separated by organic ligands, which define multiple-quantum-well electronic structures while also influencing the octahedral symmetry through a combination of intermolecular forces[9–11]. The 2D periodicity of the lattice facilitates oscillating electron-phonon interactions[12] and coherent exciton transport[13,14]. Additionally, long-range crystallinity of these 2D-structures formed by solution-processing methods have demonstrated tunable nonlinear optical behaviors[15], ferroelectricity[16,17], and spin dynamics[18,19].

In 2D semiconductors with strong spin-orbit coupling, inversion symmetry breaking induces Rashba band splitting, wherein the spin-up and spin-down bands exhibit right-handed and left-handed circular polarized orbital momentum ($\sigma^+$ and $\sigma^-$)[20–27]. The Rashba band structures of 2D semiconductors can support ultralong-lived circularly polarized orbital momentum, leading to metastable orbital magnetic dipoles, which presents a fundamental platform to explore spin-switchable phenomena. Rashba splitting in the ferroelectric 2D-perovskite 4,4-DFPD$_2$PbI$_4$ was recently demonstrated via the circular

[1]Department of Materials Science and Engineering, University of Tennessee, Knoxville, TN, USA. [2]Neutron Scattering Division, Neutron Sciences Directorate, Oak Ridge National Laboratory, Oak Ridge, TN, USA. [3]Center for Nanophase Materials Sciences, Oak Ridge, TN, USA. [4]Department of Materials Science and Engineering, Seoul National University, Seoul, Republic of Korea. ✉e-mail: lauterv@ornl.gov; bhu@utk.edu

photogalvanic effect, revealing spin-momentum coupling and spin texture switching[16]. Rashba bands can be selectively populated by using a circularly polarized excitons with the corresponding handedness ($\sigma^+$ and $\sigma^-$)[28–30]. When the circularly polarized excitons conserve their circular polarizations within the PL lifetime window, CPL can be observed. CPL provide dual fundamental impacts. First, CPL functions as an optical analogue for simulating spin chirality, offering a convenient approach to investigate chirality-induced spin selectivity (CISS). Second, CPL carries spin information from Rashba band structures, presenting an important experimental tool for exploring spin-switchable optical effects.

Due to strong SOC accompanied by largely anisotropic intra-plane and inter-plane interactions, 2D-superlattice perovskites can demonstrate Rashba splitting to lift the spin degeneracy in band structures, generating the so-called spin-up and spin-down Rashba band structures carrying right-handed and left-handed circularly polarizable orbital momentum[31,32]. Our early studies indicate that circularly polarized photoexcitation can induce magnetization in high-quality hybrid perovskite films through a photoexcited interfacial effect with a ferromagnetic metal, leading to an optically-induced magnetization[33]. Together, these findings suggest that 4,4-DFPD$_2$PbI$_4$'s Rashba band structures can support spin-selective phenomena by the direct interaction between circularly polarized orbital order and spin order.

In this work, we utilized a refined spin-coating method with precursor refinement and optimized growth conditions, and prepared high-quality, single crystalline-like 2D-superlattice perovskite films (4,4-DFPD$_2$PbI$_4$) with well-defined periodic interlayer structures. Interfacing this 2D-superlattice with a ferromagnetic Co layer forms a multiferroic perovskite/Co interface, where circularly polarized orbital order interacts with spin order, enabling spin-selective optical phenomena within Rashba band structures. Circularly polarized photoexcitation ($\sigma^+$) of 343 nm was used to selectively populate the $\sigma^+$ band, generating circularly polarized excitons ($\sigma^+$ excitons) with right-handedness. An external magnetic field was applied parallel to the film plane and then switched between +1 T and −1 T to change the spin order in-plane between +**B** and −**B** directions. Simultaneously, the CPL were measured with $\sigma^+$ and $\sigma^-$ conditions under +1 T and −1 T, respectively, for a 2D-superlattice perovskite film prepared on a ferromagnetic cobalt layer.

## Results

The experimental results include the basic characteristics of 2D-superlattice perovskite films prepared by the spin-coating method, optically operating circularly polarizable orbital momentum within Rashba band structures, spin-switchable orbital momentum through intersystem crossing between Rashba band structures, and dynamics of spin-switchable phenomena studied by circularly polarized pump-probe transient absorption spectroscopy.

### Basic characteristics of 2D-superlattice perovskite films prepared by spin-coating method

Figure 1 shows the basic characteristics for the 2D-superlattice perovskite 4,4-DFPD$_2$PbI$_4$ ([C$_5$H$_{10}$F$_2$N]$_2$PbI$_4$) films prepared by our spin-cast method. This 2D-superlattice perovskite with Ruddlesden Popper structure type belongs to space group Aba2 with stacking of Pb-I octahedral layers confined by the 4,4-Difluoropiperidine amine

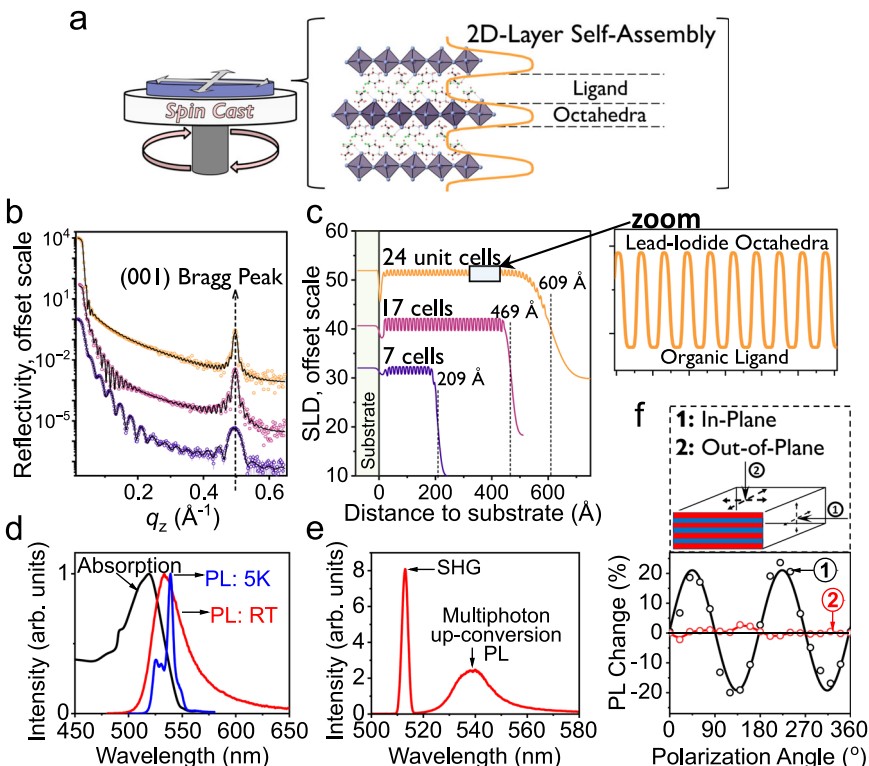

**Fig. 1 | Characterizations for 2D-superlattice perovskite film with thickness of 70 nm.** "a.u." represent arbitrary units. **a** Schematic illustration to show 2D-superlattice film prepared by spin coating. **b** Experimental X-ray reflectivity (XRR) as a function of the wavevector transfer Q (A$^{-1}$). **c** SLD fitting showing superlattice-like ordering for perovskite films with different thicknesses. Zoom-in provided of SLD periodic layered structure with alternating layers of 4,4-DFPD organic cation and Pb-I octahedra prepared by spin-cast method. **d** Optical absorption and photoluminescence spectra at room temperature and at 5 K. **e** Second harmonic generation (SHG) and up-conversion photoluminescence spectra. **f** Schematic diagram (top panel) illustrates two excitation configurations (parallel and normal to film surface) used in polarization-dependent photoluminescence measurements. Polarization-dependent photoluminescence intensity (bottom panel) changes as the exciting laser linear polarization is rotated using half-wave plate. Beam applied parallel to the film plane demonstrates a significant variation in response to the rotating polarization direction. Beam applied perpendicularly to the film, with a normal incident angle, results in negligible changes in PL intensity.

cation[34]. The X-ray reflectometry (XRR) data illustrate the superlattice structures in 2D perovskite films with different thicknesses (Fig. 1b, c). Reflectivity data display fringes corresponding to the total thickness of the film and Bragg-peaks at $Q_z = 0.5$ Å$^{-1}$ determined by the 6.2 Å period of the superlattice structure of 2D perovskite films. The depth profile of the scattering length density (SLD) obtained from the fit to the data, displayed in Fig. 1b, reveals a self-assembled periodic structure with film thicknesses of 60.9, 46.9, and 20.9 nm respectively and low roughness of ~1 nm throughout the 2D film superlattices. The bandgap of 2D-superlattice perovskite film with 70 nm thickness can be estimated to be 2.26 eV according to the optical absorption spectrum in Fig. 1d. The photoluminescence (PL) is shown as a broad spectrum with the peak of 533 nm at room temperature (Fig. 1d). At the low temperature of 5 K, the PL spectrum becomes well resolved with the primary peak at 539 nm and multiple shoulder peaks. The crystalline 2D perovskite 4,4-DFPD$_2$PbI$_4$ additionally exhibits ferroelectricity, as recently reported[34,35]. Here, the high-quality 2D-superlattice film demonstrates a strong second harmonic generation (SHG) signal at 513 nm and 2-photon up-conversion photoluminescence (UCPL) peaked at ~540 nm when excited by a 1027 nm laser, shown in Fig. 1e. Notably, the light-emitting dipoles are well oriented within 2D film plane, shown by a significant anisotropy of PL intensity when the incident photoexcitation beam is operated with its linear polarization perpendicular and parallel to 2D film plane (Fig. 1f). Together, these characteristics of the 2D-superlattice perovskite film make it a high-quality 2D platform to explore unique optical properties.

## Optically operating circularly polarizable orbital momentum within Rashba band structures

Essentially, 2D-superlattice perovskite films are formed with non-degenerate spin-polarized Rashba band structures driven by spin-orbital coupling (SOC) and symmetry breaking. The spin-up and spin-down Rashba band structures carry circularly polarized orbital momentum with right and left handedness. This leads to right-handed and left-handed circularly polarized orbitals associated with spin-up and spin-down band structures. Through momentum conservation, the spin-up and spin-down Rashba band structures carrying right-hand and left-hand circularly polarized orbital momentum can be selectively excited by right-handed and left-handed circularly polarized photoexcitations, respectively. When the spin-orbital coupling lifts the spin degeneracy in Rashba band structures, optically exciting spin-up and spin-down Rashba band structures can eventually generate different photocurrents, known as photogalvanic effect on photocurrent[27,36]. We should note that, when right/left-handed circularly polarized photoexcitation excites spin-up/down Rashba band structures to generate right/left-handed circularly polarized excitons, CPL can be observed under the condition that circularly polarized excitons can conserve their circular polarizations within photoluminescence lifetime window. This can be named as photogalvanic effect on CPL. Therefore, using spin-up and spin-down Rashba band structures carrying right-handed and left-handed circularly polarized orbital momentum presents a fundamental method to generate a CPL under circularly polarized excitation. Indeed, the CPL from 2D perovskites have been widely observed[37,38]. Especially, when 2D perovskites are prepared with chiral structures, the CPL can be largely enhanced[39–42]. However, it still demands further experimental effort to clarify and elucidate how chiral structures can affect the circularly polarized orbital momentum within Rashba band structures, enhancing the CPL. More importantly, the right-handed and left-handed circularly polarized orbitals in spin-up and spin-down Rashba band structures are considered as orbital magnetic dipoles. Theoretically, this provides a possibility to use Rashba band structures as spin filters, leading to spin-valve transport phenomena. This theoretical scenario has been experimentally verified by using chiral 2D perovskites (R/S/racemic-1-(1-naphthyl)ethylammonium lead bromide) as spin election layer in spin valve device[24]. In this study, we explore the new fundamental effects in 2D-superlattice perovskite films: spin-switchable optical phenomena within Rashba band structures through multiferroic interface (perovskite/Co) design. Figure 2 shows that right-handed ($\sigma^+$) and left-handed CPL ($\sigma^-$) can be generated by right-handed ($\sigma^+$) and left-handed ($\sigma^-$) circularly polarized photoexcitation, respectively, in the 2D-superlattice perovskite film at both 5 K and room temperature (293 K). At 5 K, the degree of polarization (DOP), given by $\frac{PL_{\sigma^+} - PL_{\sigma^-}}{PL_{\sigma^+} + PL_{\sigma^-}}$, reaches 8.1 % and 7.2 % under ($\sigma^+$) and ($\sigma^-$) excitations. At room temperature, the DOP is reduced to 0.8 % and 0.6 % under ($\sigma^+$) and ($\sigma^-$) excitations. This temperature dependence of CPL implies a strong spin-phonon coupling occurred in 2D-superlattice film. Furthermore, the PL shows a much longer lifetime (0.1 ms) at 5 K

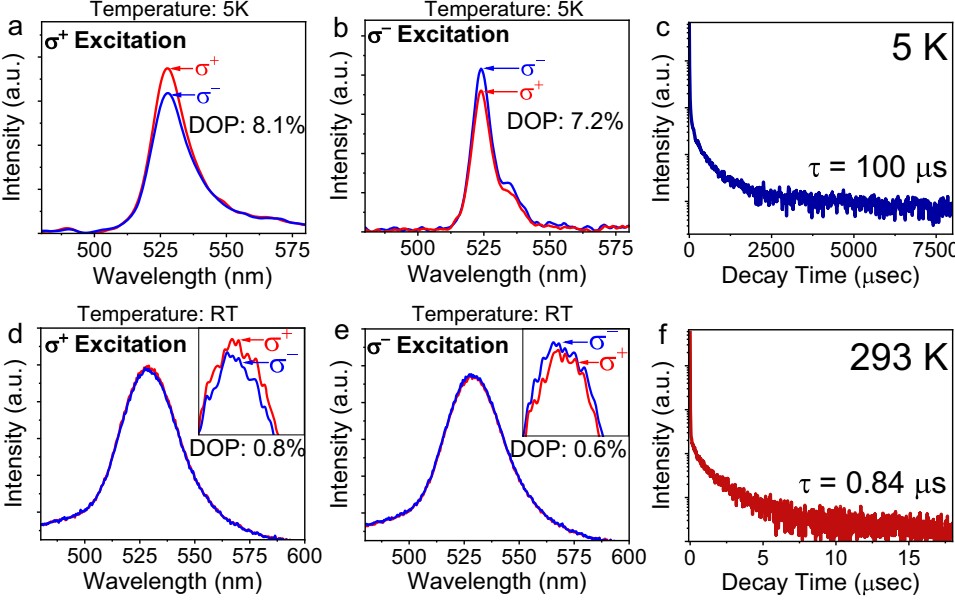

**Fig. 2 | CPL and PL lifetimes measured at 5 K and room temperature (RT).** "a.u." represent arbitrary units. **a** CPL under $\sigma^+$ excitation at 5 K. **b** CPL under $\sigma^-$ excitation at 5 K. **c** PL lifetime at 5 K. **d** CPL under $\sigma^+$ excitation at room temperature. **e** CPL under $\sigma^-$ excitation at room temperature. **f** PL lifetime at room temperature.

as compared to a much shorter lifetime (0.84 μs) at room temperature (Fig. 2c, f). Importantly, CPL can be observed only if the lifetime of circularly polarized orbital polarizations is overlapped with the PL lifetime. Here, the observed CPL provides a key indication: within Rashba band structures in the 2D-superlattice perovskite film, circularly polarized orbitals exhibit an exceptionally long time constant, reaching 0.1 ms at 5 K and 0.84 μs at room temperature.

## Spin-switchable circularly polarizable orbital momentum through intersystem crossing between Rashba band structures

Now, we explore spin-switchable phenomena in Rashba band structures by using multiferroic perovskite/Co interface design based on magnetic field effects of CPL (Fig. 3a). The ferromagnetic Co film with a thickness of 17 nm was deposited on the 2D-superlattice perovskite film. The Co layer was then covered by gold (Au) layer of 12 nm. The combined Au/Co layers were partially transparent to allow the optical transmission for both laser excitation beam and PL signal. The ferromagnetic spins on the Co surface are switched between +1 **T** and −1**T** directions while circularly polarized photoexcitation beam is applied to the 2D-superlattice film at the incident angle of 20° to the normal direction within allowable spatial arrangement. Interestingly, we can see that changing the spin order between +1 **T** and −1 **T** can switch the CPL between σ⁺ and σ⁻ handedness in the 2D-superlattice perovskite film prepared with Co layer at room temperature (Fig. 3b, c). Specifically, the spin order at +1 **T** direction largely corresponds to the σ⁺ CPL with the DOP boosted to 2.1% in the 2D-superlattice/Co film at room temperature under right-handed circularly polarized photoexcitation through perovskite/Co interface (Fig. 3b). When the spin order on the Co surface is changed to −1**T** direction, the CPL is surprisingly switched from σ⁺ handedness to σ⁻ handedness, shown as the σ⁻ CPL with the DOP of −3.1% still under right-handed circularly polarized photoexcitation (Fig. 3c). This presents a unique spin-switchable phenomenon in 2D-superlattice film at room temperature. We should also note that this spin-switchable phenomenon requires that the σ⁺ circularly polarized excitons populated in spin-up band undergo an intersystem crossing in momentum space to transfer to the spin-down band and convert into the σ⁻ circularly polarized excitons within Rashba band structures, enabled by switching the spin order on the Co surface. This

spin-enabled intersystem crossing must satisfy the momentum conservation, given by Eq. (1)

$$M_{\sigma^+} - S = M_{\sigma^-} \tag{1}$$

where $M_{\sigma^+}$ and $M_{\sigma^-}$ are the orbital momentum in spin-up and spin-down Rashba bands in 2D-superlattice film. $S$ is the spin momentum on ferromagnetic Co surface. Clearly, the spin-orbit interaction becomes the necessary condition to enable spin-switchable phenomena in 2D-superlattice perovskite film through momentum conservation in Rashba band structures.

## Dynamics of spin-switchable phenomena studied by circularly polarized pump-probe transient absorption spectroscopy

The dynamics of the spin-switchable phenomena were explored using circularly polarized transient absorption spectroscopy (TAS) under positive and negative magnetic fields. Circularly co-polarized and cross-polarized pump (343 nm) and probe beams are used while the magnetic field (+1 **T** and −1 **T**) is applied during measurements. Figure 4a illustrates the experimental setup where circularly polarized pump and probe beams were applied through partially transparent Au/Co layer to 2D-superlattice perovskite film, while magnetic field was oriented parallel to the film plane. The transient absorption spectrum of the heterostructure film has a strong photoinduced absorption (PIA) at ~525 nm, shown in Fig. 4b. The 343 nm pump magnifies hot-carrier cooling, during which time the carriers red-shift from 520 nm to the 525 nm band edge. In order to monitor the PIA spin dynamics, varying conditions of σ⁺ and σ⁻ polarized pump beams, σ⁺ and σ⁻ polarized probe beams, and positive and negative magnetic field directions are applied. The circularly polarized transient absorption ($\sigma_{probe}^+ - \sigma_{probe}^-$) is determined by the intensity difference of right-handed and left-handed circularly polarized probe signals at ~525 nm as a function of delay time. This differential spin-polarized dynamics under both σ⁺ and σ⁻ pump conditions are shown in Fig. 4c. When pumping with σ⁺ polarization, a positive magnetic field (+1 **T**) promotes σ⁺ polarized state, while a negative magnetic field (−1 **T**) induces a prompt switch to the σ- polarized state. The onset of measurable spin polarization is rapid, such that the transition between circular polarized bands occurs

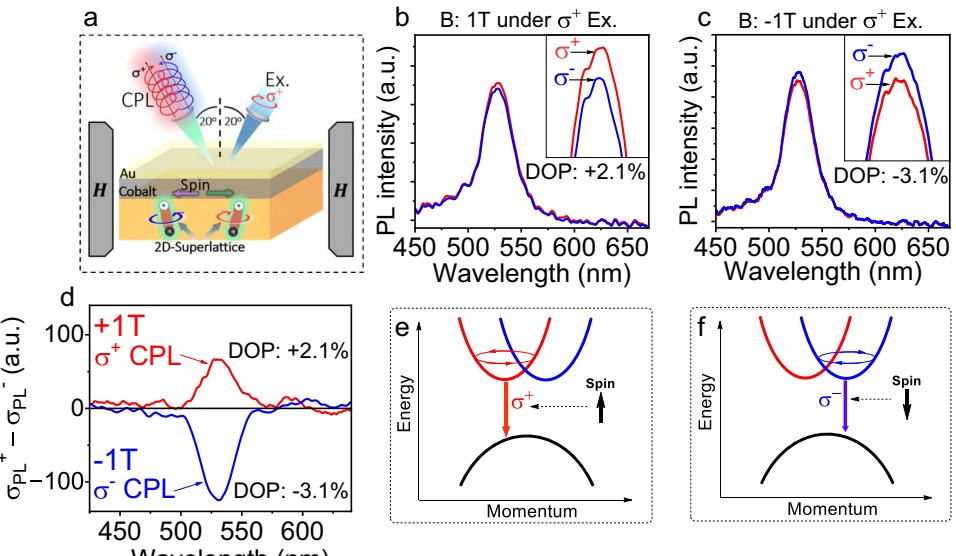

**Fig. 3 | Spin-switchable CPL between σ⁺ and σ⁻ handedness within Rashba band structures by switching spins on Co surface between +1 and -1T directions at room temperature.** "a.u." represent arbitrary units. **a** Schematic showing the experimental design of spin-switchable CPL measurement. **b** CPL at +1 T. **c** CPL at -1T. **d** Subtracted spectra (σ⁺-σ⁻) to show spin-switchable CPL enabled by switching spin order between +1 T and −1T. **e** Schematic diagram for σ⁺ CPL within Rashba band structures enabled by spin-up order on Co surface. **f** Schematic diagram for σ⁻ CPL within Rashba band structures enabled by spin-down order on Co surface.

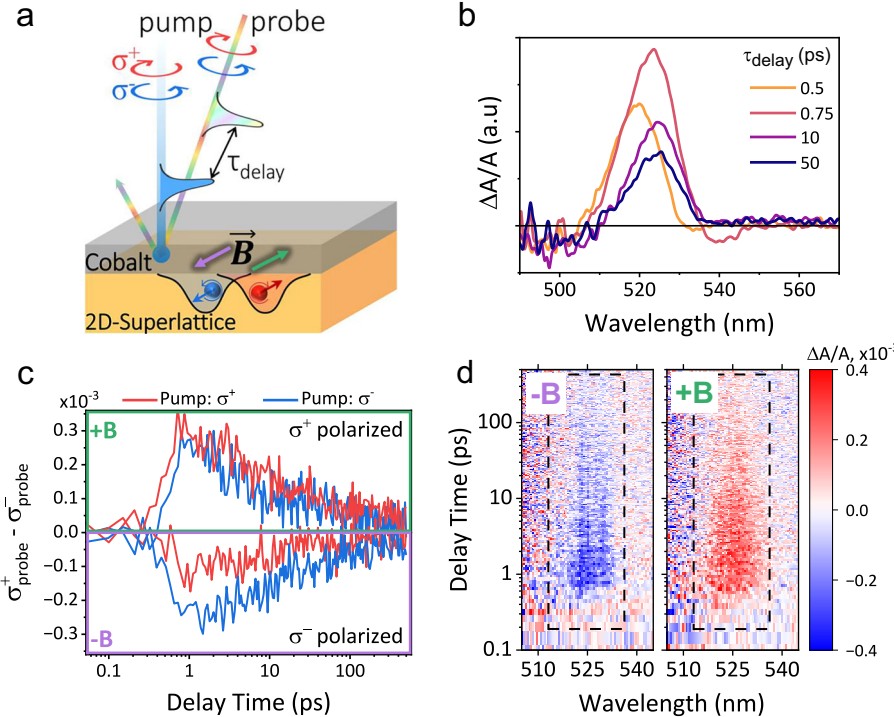

**Fig. 4 | Dynamics of spin-switchable phenomena in 2D-superlattice perovskite film interfaced with Co layer shown by transient absorption spectroscopy under applied B-field.** "a.u." represent arbitrary units. **a** Schematic representation of the polarized pump-probe setup, highlighting the spin-polarized excitation and detection on the 2D-superlattice interfaced with Cobalt layer. The variable time delay (τ) between pump and probe pulses monitors dynamics of spin polarized states within the 2D-superlattice, while under applied external magnetic field (+1 **T** or −1 **T**). **b** The transient absorption spectra shown at varying delay times under σ⁺ pump and σ⁺ probe. **c** Differential spin-polarized dynamics under two distinct pump conditions (σ⁺ and σ⁻) and two magnetic field orientations (+**B** and −**B**). The curves represent the difference between σ⁺ and σ⁻ polarized probes, i.e., time-resolved spin polarization, with +**B** resulting in σ⁺ polarized state and −**B** resulting in σ⁻ polarized state. **d** 2D heat maps of the differential spin response, shown for both magnetic field polarities under σ⁻ polarized pump. Color intensity represents the magnitude of differential absorption (ΔA/A). In the presence of a positive magnetic field the dominant positive (red) response signifies σ⁺ spin polarized excited states, while a negative magnetic field with dominant negative (blue) response signifies σ⁻ spin polarized excited states.

during hot-carrier cooling. The dynamic results are constant across the energy landscape seen in 2D heat maps shown in Fig. 4d. The red and blue color indicating σ⁺ and σ⁻ spin polarized states, respectively, have a gradual intensity decay within the polarized states monitored under the PIA. This circularly polarized pump-probe transient absorption indicates three critical points. First, the positive and negative circularly polarized transient absorption (σ_probe⁺ - σ_probe⁻) represent the population dominated on the σ⁺ and σ⁻ Rashba band structures when switching magnetic field between +1 **T** and −1 **T**. This presents clear experimental evidence to confirm that altering the direction of spin order in the Co layer can directly switch the population of photo-excited carriers between spin-up and spin-down Rashba band structures, consequently switching the orbital order of light-emitting excitons between σ⁺ and σ⁻ polarizations in 2D-superlattice perovskite film. Second, at positive spin order under +1 **T**, right-handed circularly polarized pump beam can quickly establish a dominant population in σ⁺ band in about 1 ps. Similarly, at negative spin order under −1 **T**, left-handed circularly polarized pump beam can also quickly establish a dominant population on σ band in - 1 ps. Interestingly, at positive/negative spin order under +1/−1 **T**, left/right-handed circularly polarized pump beam, which has the opposite handedness to the circular orbital momentum of Rashba band structures, can still establish a dominant population in σ⁺/σ⁻ band. This observation reveals that spin-switchable phenomena are enabled by intersystem crossing between σ⁺ and σ⁻ Rashba bands in 2D-superlattice perovskite film upon switching the spin order in the Co substrate. The opposite-handedness circularly polarized σ⁻/σ⁺ pump beam exhibits a very small delay of 0.1 ps in rising time, as compared to the same-handedness circularly

polarized beam (σ⁺), towards establishing a dominant population in the Rashba band (σ⁺/σ⁻ band) cooperative with the spin order (+1/−1 **T**). This reveals that the intersystem crossing between σ⁺ and σ⁻ Rashba bands in 2D-superlattice perovskite film is an ultrafast process with the time constant of 0.1 ps induced by switching the spin order between positive and negative directions in the Co substrate. Third, the orbital orders established in σ⁺ and σ⁻ Rashba bands are slowly relaxing and remaining at 0.1 ns under the spin orders at both positive and negative directions, providing the necessary condition to enable spin-switchable CPL in 2D-superlattice perovskite film interfaced with Co layer.

## Discussion

Essentially, the spin-switchable phenomena require a long-range interaction between the orbital order of 2D-superlattice perovskite film and the spin order of Co layer through the multiferroic perovskite/ Co interface design. To elucidate the interaction between the orbital order and spin order, we applied polarized neutron reflectometry (PNR) combined with *in operando* circularly polarized photoexcitation. The PNR is a highly penetrating depth-sensitive technique to probe the depth-spatial profiles with a resolution of 0.5 nm for chemical and magnetic components. By detecting the reflected spin-polarized neutrons, the PNR studies can provide the detailed information for optically activated spin dipoles with spatial profile in perovskite films[33]. The depth profiles of the nuclear and magnetic scattering length densities (NSLD and MSLD) correspond to the depth-spatial distributions of the chemical elements and in-plane magnetization on atomic scale, respectively[43–45]. PNR is powerful in

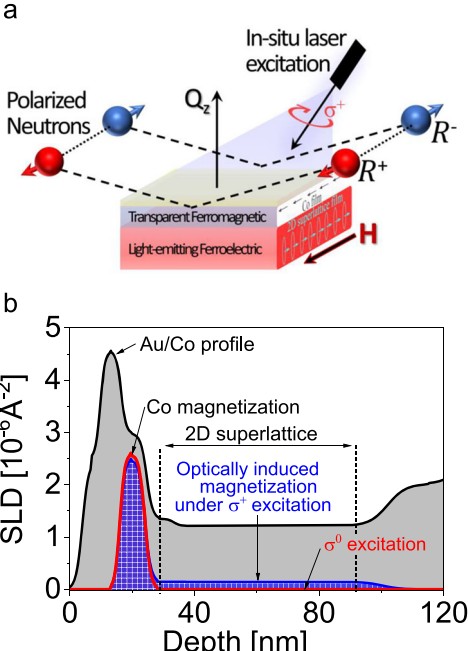

a

b

**Fig. 5 | Results from probing magnetism in 2D-superlattice perovskite film with Co surface by using polarized neutron reflectometry (PNR) experiment with in-situ polarized photoexcitation. a** Schematic illustration of the ferromagnetic/ferroelectric thin film heterostructure film as the subject of the spin-polarized neutron reflectometry experiment with polarized photoexcitation (405 nm) and under the applied in-plane magnetic field. **b** Summarized magnetic and nuclear scattering length density (SLD) for the PNR measurement conducted at 10 K. The black line corresponds to the nuclear SLD profile for Co layer covered with Au (labeled between 0 and 29 nm in x-axis) deposited on 2D-superlattice perovskite film (labeled between 29 nm and 89 nm in x-axis). Blue and red lines correspond to the magnetic scattering length density. The blue line is for the magnetism from Co in x-scale between 12 nm and 29 nm, and the optically induced magnetism in 2D-superlattice perovskite film (labeled between 29 nm and 89 nm in x-axis) under circularly polarized 405 nm laser beam ($\sigma^+$ excitation). The red line is the magnetism under linearly polarized 405 nm laser beam ($\sigma^0$ excitation).

simultaneously and nondestructively characterize chemical and magnetic natures of buried interfaces[46]. Here, we observe optically induced magnetization up to 70 nm in the entire 2D-superlattice perovskite film interfaced with a Co layer under circularly polarized photoexcitation, leading to a super long-range spin-orbit interaction (Fig. 5). In contrast, optically induced magnetization disappears when linearly polarized photoexcitation ($\sigma^0 = \sigma^+$ and $\sigma^-$) is applied to excite both $\sigma^+$ and $\sigma^-$ bands towards cancelling the orbital order within Rashba band structures. This verifies that the selective interaction between the orbital order in 2D-superlattice perovskite film and the spin order in Co layer can be established through multiferroic perovskite/Co interface design to generate an optically induced magnetization. Clearly, the super long-range interaction between spin order and orbital order provides a critical mechanism to enable spin-switchable phenomena in 2D-superlattice perovskite films through Rashba band structures.

To further understand the spin-switchable phenomena between spin-up and spin-down Rashba band structures, we measured magnetic field effects of CPL by applying circularly polarized photoexcitation applied through 2D-superlattice perovskite film surface in the multiferroic perovskite/Co sample. Exciting through the 2D-superlattice surface minimizes the spin ordering effects from the Co surface, allowing the effects of magnetic field to be detected. When the $\sigma^+$ CPL is generated with the DOP of 1.0% in the 2D-superlattice perovskite film without magnetic field by right-handed circularly

polarized laser beam at room temperature, applying +**B** field changes the CPL from $\sigma^+$ handedness to $\sigma^-$ handedness with the DOP of −1.4% (Fig. 6a). Surprisingly, applying -**B** field still changes the CPL from $\sigma^+$ handedness to $\sigma^-$ handedness with the DOP of −1.2% (Fig. 6b). Clearly, applying an external magnetic field, no matter at positive or negative direction, always induces an intersystem crossing from the excited $\sigma^+$ band to the un-excited $\sigma^-$ band within Rashba structures. This provides a critical understanding that the intersystem crossing between $\sigma^+$ band to the un-populated $\sigma^-$ band within Rashba structures is a spin flipping process, and an external magnetic field can introduce such spin flipping to convert the circularly polarized excitons from excited band to un-excited band, leading to magnetic field-induced intersystem crossing in Rashba band structures. Specifically, the excited Rashba band is largely populated with circularly polarized excitons, leading to a high entropy in population as compared to un-excited Rashba band. When spin flipping is allowed, this entropy difference intends to shift the circularly polarized excitons populated in the excited $\sigma^+$ band to the unexcited $\sigma^-$ band within Rashba structures, generating an intersystem crossing. However, shifting circularly polarized excitons from $\sigma^+$ band to $\sigma^-$ band requires both energy and momentum conservations. In general, the momentum conservation determines whether an intersystem crossing can occur between $\sigma^+$ to $\sigma^-$ bands, while the energy difference governs the rate of intersystem crossing. Normally, momentum conservation through spin flipping plays an important role in determining the intersystem crossing, since the energy conservation can be conveniently satisfied by lattice vibrations through electron-phonon coupling mechanism. The application of an external magnetic field can introduce spin scattering into circularly photoexcited excitons and thus increases the intersystem crossing from excited band to unexcited band within Rashba band structures, consequently switching the CPL from $\sigma^+$ to $\sigma^-$ (Fig. 6d).

We note that magnetic field effects provide an important experimental method to elucidate the origin of CPL. In general, CPL phenomena can be often observed with a high degree of circular polarization through birefringence effects in crystalline structures formed with anisotropic morphologies. The birefringence effects can inevitably cause a phase difference between slow and fast optical axis in crystalline structures, leading to the wave vector of light output with elliptically rotating feature shown as CPL phenomena. This birefringence-induced CPL become a challenging issue to distinguish from orbital order-induced CPL. However, the birefringence-induced CPL lack magnetic field effects. Therefore, magnetic field effects present a unique tool to elucidate the origin of CPL.

In summary, our studies of CPL in a magnetic field indicate that the circularly polarized $\sigma^+$ and $\sigma^-$ orbitals within Rashba band structures in 2D-superlattice perovskite can selectively interact with the spin order at the ferromagnetic surface through multiferroic perovskite/Co interface design. The spin order at +**B** direction can largely increase the $\sigma^+$ CPL with the DOP boosted from 0.3% to 2.1% at room temperature under right-handed circularly polarized photoexcitation. This indicates that the ferromagnetic spins reduce the de-phasing of circularly polarized orbitals in the excited Rashba band, consequently increasing the CPL. Very interestingly, when the spin order is switched from +**B** to −**B**, the CPL changes from $\sigma^+$ and $\sigma^-$ handedness with the DOP altered from 2.1% to −3.1%. This presents a unique spin-switchable phenomenon in Rashba band structures in 2D-superlattice perovskite film. Essentially, this spin-switchable phenomenon reflects that the selective interaction is occurred between the circularly polarized orbitals in Rashba band structures and the ferromagnetic spins on the Co surface through multiferroic perovskite/Co interface design. Specifically, the $\sigma^+/\sigma^-$ circularly polarized orbitals in Rashba band structures are interacting with the ferromagnetic spins at +/− **B** direction on the Co surface. This selective interaction between orbital order and spin order is indeed a magnetization process, shown by polarized neutron reflectometry. When this selective interaction enables spin-

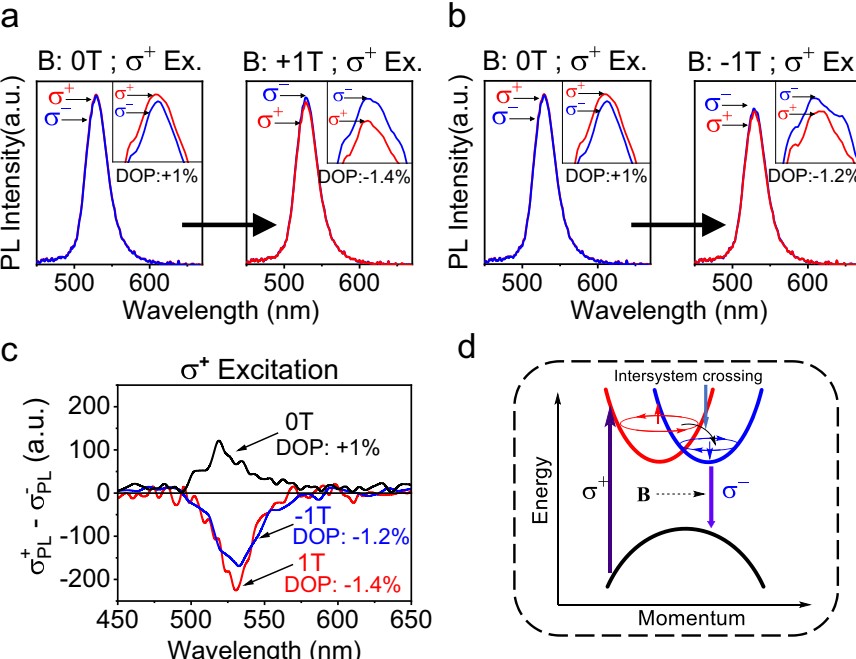

**Fig. 6 | Magnetic field effects of CPL by applying σ⁺ excitation through perovskite surface in 2D-superlattice perovskite film (70 nm) prepared on Co surface.** "a.u." represent arbitrary units. **a** Magnetic field at +**B** changes σ⁺ CPL to σ⁻ CPL. **b** Magnetic field at −**B** still changes σ⁺ CPL to σ⁻ CPL. **c** Subtracted spectra (σ⁺- σ⁻) to show magnetic field effects: field (regardless +**B** and −**B**) always changes σ⁺ CPL to σ⁻ CPL. **d** Schematic diagram to illustrate intersystem crossing from excited band to un-excited band within Rashba structures induced by magnetic field through spin flipping.

switchable phenomena, it requires an intersystem crossing in momentum space between spin-up band carrying the **σ⁺** circularly polarized orbital momentum and the spin-down band carrying the **σ⁻** circularly polarized orbital momentum in 2D-superlattice perovskite film. When removing spin ordering effect by exciting 2D-superlattice film through perovskite surface in multiferroic interface design, we find that an external magnetic field induces an intersystem crossing always from excited band to unexcited band within Rashba band structures in 2D-superlattice perovskite film. This indicates that the intersystem crossing between spin-up and spin-down Rashba band structures is a spin flipping process, leading to magnetic field-dependent CPL within Rashba band structures in 2D-superlattice perovskite film. Clearly, spin switchable optical phenomena can be realized in the 2D-superlattice perovskite through spin-operative momentum symmetry breaking between **σ⁺** and **σ⁻** bands within Rashba band structures.

## Methods
### Thin film preparation
Millimeter-sized crystal shards of 4,4-DFPD$_2$PbI$_4$ are grown by precipitation during slow cooling of precursor solution. Difluoropiperidine is added to solution of 10 mL of hydroiodic acid (57%, unstabilized) with hypophosphoric acid (~1 mL or until hydoiodic acid turns a pale yellow) at 0.1 M, then stirred at 50 °C for 30 mins. Then lead iodide (99.999%) is added to the solution at a stoichiometric ratio (0.05 M) and the solution is further stirred at 100 °C until fully dissolved appearing as a pale-yellow solution. During cooling from 100 °C to 60 °C at a rate of 1 °C/hr, the crystals begin to nucleate at ~85 °C and are harvested at 60 °C. The harvested crystals are washed and stirred in anhydrous methyl acetate three times then dried at 80 °C under vacuum for 5 h. Typical yield of growth is > 80% and act as the high purity precursors for spin-cast solution preparation. To prepare 2D-superlattice films, the dried 4,4-DFPD$_2$PbI$_4$ crystals are dissolved in dimethyl sulfoxide at ratios of 0.1 g/mL and 0.2 g/mL, for 25 nm and 80 nm thicknesses, and filtered using a 0.2 µm pore-size PTFE filter. Spin casting takes on a 4-step process where we use two spin speeds to optimize uniformity of the solution across the substrate, then put under vacuum to stimulate nucleation, and finally crystallize by thermal annealing. Spin casting to produce crystalline-like films is optimized as a two-step process of (1) 500 rpm for 7 s (2) 8000 rpm for 60 s. Immediately after spin-coating the films are vacuumed for 6 mins and annealed at 100 °C for 10 min, producing a bright, fluorescent orange color film. These solution processing methods, crystal growth and thin film fabrication process, are illustrated in Fig. S6. The fluorescent-color appearance is found to be an obvious characteristic of films with long spin-lifetimes and high crystallinity. These steps produce crystalline-like films. The samples were characterized by XRR, which revealed an excellent self-assembled 2D superlattice structure and a low surface roughness of around 1 nm across the entire 2 × 2 cm² sample. Noticeably, 2D-perovskite crystalline grains in the spin-cast films prefer to crystallize with the [001] direction normal to the substrate such that the octahedral layers are parallel to the substrate surface, although near perfect orientation has been achieved in both vertical and lateral directions through engineering the spin-coating process[47,48]. Therefore, a spin-cast method is used to develop 2D-superlattice films, featuring a crystalline-like organization of (001) planes oriented parallel to the substrate surface, with high uniformity across a 2 × 2 cm² area and a tunable thickness between 10 nm and 80 nm. Finally, on top of the 4,4-DFPD$_2$PbI$_4$ film prepared on glass substrate, partially transparent layers of cobalt (17 nm) and gold (12 nm) are thermally evaporated.

### Polarized light emission
Without polarization resolved, the temperature-dependent photoluminescence, time-resolved photoluminescence lifetime, and second harmonic generation, were measured using a Horiba Fluorolog 3 spectrometer and PPD-900 photon counting detector. The excitation sources of 343 nm (photoluminescence) and 1027 nm (second

harmonic generation) are the filtered harmonics generated from a 290 fs pulsed laser (Pharos laser, Light Conversion). Polarization resolved photoluminescence was measured using Ocean Optics Flame spectrometer. The heterostructure film was excited through the cobalt layer at a 20° incident angle and photoluminescence is collected and collimated with −20° angle from normal, as depicted in Fig. 3. With an experimental geometry using an incident angle of 20°, the interaction between the circularly polarized light and the Co layer induces optically generated magnetizations whose magnitudes depend on the direction of the applied magnetic field (Fig. S7). The 343 nm excitation is polarized first through a crystal linear polarizer, followed by a zero-order $\lambda/4$ waveplate mounted on a rotation stage. The collimated emission passes through a superachromatic $\lambda/4$ waveplate mounted on a rotation stage, followed by a crystal linear polarizer, before entering the spectrometer. Peak photoluminescence intensity is taken for calculation of degree of polarization. Low temperature photoluminescence measurements were conducted in a closed cycle helium cryostat with optical windows, reaching a minimum temperature of 5 K. The observed shoulder on the high-energy side of the PL spectra in certain figures originates from glass, especially pronounced when the PL signal is weaker due to conditions such as the presence of Co/Au layers or upon glass encapsulation (Fig. S8). Cobalt was magnetized in-plane using 1 T and −1T fields applied by an electromagnet. The field direction and magnitude were constant during the spin-operable measurements. At ±1 T, magnetic saturation of the cobalt layer provides a uniform magnetic field that isolates the perovskite's intrinsic spin-state transitions by minimizing the influence of domain wall movements and other complexities introduced by magnetic domain structures near the coercive field in the amorphous cobalt film.

### Polarized pump-probe transient absorption spectroscopy

Transient absorption measurements were performed using a Helios fire spectrometer (Ultrafast Systems LLC), 343 nm pump beam operating at 1 kHz, and white light probe beam. The probe beam was generated by directing a 1027 nm laser through a delay line and then through a $CaF_2$ crystal, producing a continuum source (480–900 nm). Transient absorption was conducted in reflection mode with the pump laser incident normal to the film and the probe beam reflecting at ~10° from normal incidence. Prior to incidence, the pump beam was aligned through a crystal linear polarizer and a zero-order $\lambda/4$ waveplate, while the probe beam passed through a crystal linear polarizer followed by a superachromatic $\lambda/4$ waveplate. Spin polarization is calculated by subtracting the transient response from the $\sigma^+$ probe from the $\sigma^-$ probe, effectively isolating the spin-dependent component of the signal. Lifetimes and dynamics of the respective spin-polarized states are assessed by comparing the decay curves from $\sigma^-$ and $\sigma^+$ pump excitations.

### X-Ray Reflectometry (XRR)

XRR measurements were performed at the Center for Nanophase Materials Sciences (CNMS), Oak Ridge National Laboratory. XRR measurements were conducted on a PANalytical X'Pert Pro MRD equipped with hybrid monochromator and Xe proportional counter. For the XRR measurements, the X-ray beam was generated at 45 kV/40 mA, and the X-ray beam wavelength after the hybrid mirror was $\lambda = 1.5406$ Å (Cu Kα1 radiation). The step size (Δ 2θ) was 0.01° and the exposure time at each step was 10 s. Crystal visualization was produced using the VESTA program and reported crystallographic reference file[34,49].

### Polarized neutron reflectometry (PNR)

PNR experiments were performed on the Magnetism Reflectometer at the Spallation Neutron Source at Oak Ridge National Laboratory[50], using neutrons with wavelengths $\lambda$ in a band of 0.2 – 0.8 nm and a high polarization of 98.5–99%. Polarization analysis of

the reflected and scattered beam can be performed using in situ polarized $^3$He system[51,52] and a supermirror analyzer[53]. Measurements were conducted in a closed cycle refrigerator (Advanced Research System) equipped with a 1.15 T Bruker electromagnet. Using the time-of-flight method, a collimated polychromatic beam of polarized neutrons with the wavelength band Δλ impinges on the film at a grazing angle θ, interacting with atomic nuclei and the spins of unpaired electrons. The reflected intensity $R^+$ and $R^-$ are measured as a function of momentum transfer, $Q = 4\pi\sin(\theta)/\lambda$, with the neutron spin parallel (+) or antiparallel (−), respectively, to the applied field. To separate the nuclear from the magnetic scattering, the spin asymmetry ratio SA = $(R^+ - R^-)/(R^+ + R^-)$ is calculated, for which SA = 0 designating no magnetic moment in the system. PNR is a highly penetrating depth-sensitive technique to probe the chemical and magnetic depth profiles with a resolution of 0.5 nm. The depth profiles of the nuclear scattering length density (NSLD) and Magnetic scattering length density (MSLD) correspond to the depth profile of the chemical and IP magnetization vector distributions on the atomic scale, respectively[46,54,55]. Based on these neutron scattering merits, PNR was applied to characterize chemical and magnetic nature of buried interfaces simultaneously and nondestructively[56]. The thickness of each individual constituent layer, the interfacial roughness, and the magnetization depth profile was obtained in detail for all samples (See Fig. 5). The SLD of the perovskite film (which measured chemical composition and density of the film) shows uniform distribution confirming the excellent quality of the film. The roughness at the perovskite/ Co and Co/Au interfaces is 3.5 nm and 2.5 nm respectively.

## Data availability

The data that support the findings of this study are available from the corresponding author upon reasonable request.

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

## Acknowledgements

The experimental studies were performed by using the time-resolved magneto-optical instrumentations supported by DURIP funding (FA9550-18-1-0472 Bin Hu) at the Institute of Advanced Materials and Manufacturing at the University of Tennessee. This research used the resources of the Center for Nanophase Materials Sciences (CNMS) under CNMS user program and Neutron Scattering Division (NSD). CNMS and NSD are DOE Office of Science User Facilities. This manuscript has been

co-authored (Oak Ridge National Laboratory: Valeria Lauter, Jong Kuem, Haile Ambaye) by UT-Battelle, LLC, under contract DE-AC05-00OR22725 with the US Department of Energy (DOE). The United States Government retains, and the publisher, by accepting the article for publication, acknowledges that the United States Government retains a non-exclusive, paid-up, irrevocable, worldwide license to publish or reproduce the published form of this manuscript, or allow others to do so, for United States Government purposes. The Department of Energy will provide public access to these results of federally sponsored research in accordance with the DOE Public Access Plan (http://energy.gov/downloads/doepublic-access-plan).

## Author contributions

B. Dryzhakov, Y. Tang, J. Keum, H. Ambaye, V. Lauter, and B. Hu designed, conducted, and analyzed the experiments. J. Kim and T. Lee discussed the experimental results. B. Dryzhakov, Y. Tang, V. Lauter, and B. Hu co-wrote the manuscript with contributions from all authors.

## Competing interests

The authors declare no competing interests.
