## [Transparent Peer Review file · Nature Communications]

Spin Switchable Optical Phenomena in Rashba Band Structures through Intersystem Crossing in Momentum Space in Solution-Processing 2D-Superlattice Perovskite Film

Corresponding Author: Professor Bin Hu

Version 0:

Reviewer comments:

Reviewer #1

(Remarks to the Author)

The manuscript titled "Spin Switchable Optical Phenomena in Rashba Band Structures through Intersystem Crossing in Momentum Space in Solution-Processing 2D-Superlattice Perovskite Film" by Bogdan Dryzhakov et al. reported the spin switchable phenomena in the multiferroic perovskite/Co interface, where the circularly polarized luminescence ($\sigma+$ and $\sigma-$) could be switched via changing the magnetic field direction (+B and -B). The momentum symmetry breaking of the Rashba band structure in this system with long orbital polarization lifetime and long-range spin-orbit interaction, provides a possible approach for developing potential spin devices. However, the quality of this manuscript does not meet the high standards of Nature communication. The followings are the main issues about this manuscript.

(1) I have doubts about the data in this manuscript. Specifically, what is the DOP of Figure 3c? The ones in both the text and Figure 3d are 3.1%, while that in Figure 3c is 3.3%. Figure 5 also has this problem, and the values stated in the main text and the figures are different.

(2) Figure 3a does not clearly express the experimental design process and there are redundant lines.

(3) The testing temperature for other data in the article is 5K, while Figure 4 shows 10K, which may have some differences. For the completeness and reliability of the data, it is recommended to retest at a temperature of 5K.

(4) A small question is raised here. We have previously tested a 10nm Au layer, but it seems that it is not semi-transparent. Are 15nm Au and 17nm Co sure to be semi-transparent? Is there optical photo that can be added to the supplement information to prove?

(5) Line 194, 20 degrees instead of 20 centigrade.

(6) Line 267, Figure 4d is not visible in the figure, it should be Figure 5d.

(7) Finally, the main problem of this work is that the degree of polarization is so low that can be considered within the error range, thereby repetitive experiments are strongly necessary to verify the existence of such polarization.

Reviewer #2

(Remarks to the Author)

This work describes a study of magnetic field control of circularly polarized photoluminescence in two-dimensional hybrid perovskites/ferromagnetic heterostructures. Beyond the ordinary circularly polarized photoluminescence that has been widely reported in hybrid perovskite thin films upon circularly polarized light illumination (Fig. 2), the highlighted work stated by the authors is that when the perovskite thin film is interfaced with the ferromagnetic, Co layer, the circular polarization of photoluminescence can be altered upon the direction of an applied external magnetic field. The reversal of circular

polarization has been attributed to the intersystem crossing between spin-polarized sub-bands caused by the Rashba splitting in hybrid perovskite films. Such a switchable behavior can also persist up to room temperature which is also concluded by the presence of super long-range spin-orbit interactions. However, the main conclusion drawn from this manuscript is not validated since this phenomenon can be simply described by a general magneto-optical Kerr effect that has been well accepted elsewhere. The observed phenomenon has nothing to do with the proposed Rashba state (no direct evidence for this statement either) and the spin-flip mechanism between spin sub-bands. Since this is the key statement in this manuscript, I cannot recommend the publication of this work in Nature Communications.

(1) While the reviewer appreciates that a superlattice-like high-quality 2D hybrid perovskite thin film is demonstrated, the authors made an incorrect statement that the observation of circularly polarized photoluminescence implies the presence of the Rashba effect in their thin films. It is not always valid. A strong spin-orbit coupling in hybrid perovskite will solely generate a J state in the CB that can be responsible for the selective circularly polarized light absorption and circularly polarized photoluminescence, similar to that in the prototypical GaAs system. No Rashba state is necessarily needed to enable this phenomenon. Please see Odenthal et al., *Nat. Phys.* 13, 894–899 (2017). A cross-check that can unambiguously prove the Rashba effect needs to be presented, such as indirect band transition probed by ultrafast PA spectroscopy or circular photogalvanic effect, etc.

(2) The field control of circular polarization is mistakenly interrupted. The observed signals would be simply generated by the magneto-optical Kerr effect (MOKE) from the cobalt at the Co/perovskite interface. There are two components in the MOKE, one real part is about the rotation of linear polarization, i.e., Kerr rotation, and one imaginary part is about a change of the ellipticity, i.e., Kerr ellipticity. Please see Qiu & Bader, *Rev. Sci. Instrum.* 71, 1243–1255 (2000). The amplitude of Kerr ellipticity changes as a function of wavelength, following the magnetic hysteresis loop of the ferromagnetic layer. In this work authors only reported the results at two opposite magnetic fields. You may imagine that if a full loop of circular polarization is measured, it will follow exactly the magnetic hysteresis of the Cobalt layer.

A similar concept has been also widely employed in the field of 2D magnets to characterize their magnetic properties using reflectance magneto-circular dichroism (RMCD) microscopy, see Huang et al., *Nature Nano.* 13, 544–548 (2018).

Please note that the authors used a typical longitudinal MOKE/RMCD technique by which the PL is collected at the incident angle of 20 degrees to the normal plane. If the incident angle is bigger, a larger MOKE/RMCD signal would be obtained. There is irrelevant to the proposed Rashba effect in hybrid perovskite thin films that the authors stated.

(3) Temperature-dependent data can be also explained by the MOKE response due to the reduced magnetization of the Co layer at 300K. By conducting a full magnetic hysteresis of the loop, the authors should observe a narrower loop because of the reduced coercive field in the Co film.

(4) Time-dependent PL results in Fig. 2 don't support the conclusion for the intersystem crossing at the Rashba effect. The authors need to conduct the time-dependent PL measurement also in their heterostructure devices. If there is any intersystem crossing or spin flip between spin-polarized subbands that dominate this process, the circular polarization of PL emission would change as a function of time at opposite field directions.

(5) The authors need to perform more control experiments to underpin the interfacial effect between the Co and hybrid perovskite thin films. One control experiment is to insert a thin Cu space layer (~2-5nm) between Co and hybrid perovskite to identify whether there is an emergent pseudo magnetic moment by magnetic proximity effect, transferring of the orbital angular moment, or direct exchange coupling.

Reviewer #3

(Remarks to the Author)

The work proposed about spin switchable circularly luminescence is relevant to the spin optoelectronics research community and goes in the current direction of modulating this optical property in novel materials. However, to fully support the conclusions and claims, a major revision of the manuscript with additional evidence/experiments is needed before considering for publication. There are missing some important details about the methodology and control experiments. Here the critical points that should be addressed:

-In the methods section, authors mentioned that “on top of the 4,4-DFPD2PbI4 film prepared on glass substrate, semi-transparent layers of cobalt (17 nm) and gold (12 nm) are thermally evaporated.” Then, some of the sentences along the text are imprecise: Abstract - “film spin-coated on ferromagnetic cobalt (Co) substrate”; introduction - “respectively, for a 2D-superlattice perovskite film prepared on a ferromagnetic Co surface.”, and this is a key point since it explains how the system under study is made. Following this line, authors should provide experimental data proving that the Co/Au thermal deposition is not damaging the perovskite layer.

-In Figure 1e, the PL spectrum at 5K shows multipieaks on the right and left sides of the main emission peak. Authors commented about the presence of these multipieaks, but not about their origin. How do authors explain the appearance of these peaks? Usually in high quality perovskite thin films the photoluminescence (PL) spectrum at low temperatures is simply narrower than at room temperature, but it is not common to observe new features in both sides of the photoluminescence emission. Even more, the shape of the PL spectrum at 5K in Figure 1 is completely different than the one in Figure 2 a-b. The same happens for the room-temperature PL spectrum but less marked. Authors should explain the reasons why this is happening in more detail.

-In figure 1g, it is shown a “PL change in %”, but in the text authors refer to “anisotropic PL intensities”. Therefore, it is misleading. Mainly because there is no methods section explaining how the PL measurements were carried out: polarization only in the excitation path? spectrometer/detector used? In the data analysis did authors use absolute intensity or integrated area?

-Regarding Figure 3, authors comment “before applying the ferromagnetic Co layer [...] CPL-DOP is 0.3%”, but there is not data showing this as well as the CPL after Co/Au deposition at 0T magnetic field. Indeed, looking at Figure 2, this value is smaller than the 0.6/0.8% reported there. Therefore, is it a question of reproducibility among different samples? But most importantly, why did the authors carry out the experiments in Figure 3 and 5 only at room temperature and not at 5K, where, as shown in Figure 2a-b, the DOP values were already higher and therefore, more accurate results could be obtained?

-Comparing Figures 3 and 5 and most concretely the PL spectra, the shape of the PL spectra shown in panels Figure 3b-c and Figure 5a-b-left and different than the ones in Figure 5a-ac-right. It seems that there is not the same background signal, since intensity counts at 450 nm are higher than at 650 nm in the first ones and in the second both are equal.

-There is not enough detail provided in the methods for the work to be reproduced. Paragraphs regarding absorbance and PL measurements, lifetime and SHG are missing. For example, in Figure 3a, the scheme indicates a 20° angle for incident excitation and collected signal, but there is no explanation. This Reviewer supposes that it should be related to the experimental set-up, but at the same time, the incident/collection angle should affect the PL spectrum since these perovskite thin films could suffer from reabsorption effects.

-One key point in research is the reproducibility of the results. Along the text, this Reviewer is not able to know in how many samples did the authors observe the “Spin Switchable Optical Phenomena”, considering the fabrication procedure: spin coating/vacuum/annealing steps for the perovskite film preparation and thermal evaporation for “17 nm” Co and “12nm” Au deposition.

Minor issues that should be addressed:

-Please correct misspellings and wrong notations along the text, e.g. “2D-sperlattice”, “20°C”; and the format/content of some of the references.

Version 1:

Reviewer comments:

Reviewer #1

(Remarks to the Author)

My questions and concerns have been addressed by the authors. The manuscript has been satisfactorily revised. Based on the major improvement made, I would like to pass it for acceptance.

Reviewer #2

(Remarks to the Author)

The authors have made deliberate efforts to respond to all the reviewers' comments. Some of the added control experiments have partially resolved my concerns but not all, particularly for the key statement: spin-switchable magneto-CPL assisted by the proposed Rashba state. Therefore, I cannot recommend the publication of this work until my concerns are fully resolved. Below I list my unaddressed concerns following the sequence in the rebuttal letter.

Comment 1: There are indeed many literature reports about the presence of the Rashba state in layered hybrid perovskite. But this cannot stop me from asking for direct experimental evidences of the Rashba effect in the currently used 2D perovskite thin film. Other groups reported the Rashba effect in their 2D perovskite thin films with a different composition and symmetry breaking, not the current one. So please provide:

(1) DFT calculations about the spin-splitting state in the 4,4-DFPD2PbI4 film and describe the Rashba parameter in the unit of eV/Å.

(2) Experimental proof of the spin-momentum relation, e.g., circular photogalvanic effect (CPGE) in the 4,4-DFPD2PbI4/Co heterostructure. The circular polarization-dependent pump-probe spectroscopy is inadequate to prove the presence of the Rashba state. I didn't see any feature of the indirect band transition. Please show the influence of the applied magnetic field on the spin-splitting sub-bands (energy shift, density state, etc.) via CPGE, leading to the field-dependent CPL polarization.

Comment 3: Please just measure the M vs. T response in the 4,4-DFPD2PbI4/Co heterostructure using a SQUID magnetometer. The change of magnetization of the Co layer would be easily identified without involving the complicated PNR measurements.

Comment 4: The current data sets (only at +/- 1T) are inadequate to prove the authors' statement. Please systemically change the strength of the magnetic field (and near the coercive field of the Co layer) to see whether the proposed intersystem crossing truly correlates with the change of spin-state.

Comment 5: If there is indeed a unique interfacial Rashba effect (not arise from the 4,4-DFPD2PbI4 itself) formed at the

interface and modulated by the magnetic layer, did the authors conduct magneto-CPL measurement in a much thicker or thinner film? Does the polarization of field-dependent CPL remain the same? The interfacial effect should exist only within < 5-10nm near the interface.

Reviewer #3

(Remarks to the Author)

Authors have made some corrections and updates in the manuscript, but the response to some of the key questions from Reviewers is vague and not precise. Here the main points that should be addressed:

- Authors claim: the protocol to prepare the perovskite films "has become a standard process for our group" "we noticed there are small variations in the CPL-DOP values between different samples". However, authors do not provide statistics and a concrete number of samples in which they have checked the reproducibility of the CPL-DOP values observed. Indeed, how can they state that the 0.8% values are not inside the error of the measurements?
- "The weak background signal is caused by the glass encapsulation layer for our samples." Related to Figure 2. Then, authors should provide the signal from the glass as reference.
- Authors claim that the 17 nm Co/12nm Au layer is semitransparent but looking to Figure S1 at the excitation wavelength used for photoluminescence measurements, i.e. 343 nm, the transmittance is < 10%. It is really challenging to state that with this low transmittance the metallic film is semitransparent.
- In the experimental section there are no details regarding the precursors amount (mg, mL,...) used for the synthesis of the crystals. Authors should include these details.
- Authors claim: "we have manually examined that changing angle does not lead to noticeable reabsorption" referring to the 20° angle used for the measurements, but there is no new graph, even just in the rebuttal letter showing this angle-independent behavior.

Version 2:

Reviewer comments:

Reviewer #2

(Remarks to the Author)

The authors have addressed my comments. The revision is satisfying. Thus I recommend the publication of this work.

Reviewer #3

(Remarks to the Author)

The manuscript has been satisfactorily revised as it seems that the authors will not include additional experiments in this work, even if they were requested by Reviewers. It can be published as it is.

made.

Responses to Reviewers

Spin Switchable Optical Phenomena in Rashba Band Structures through Intersystem Crossing in Momentum Space in Solution-Processing 2D-Superlattice Perovskite Film

Bogdan Dryzhakov¹, Yipeng Tang¹, Jong Keum^{2,3}, Haile Ambaye², Jinwoo Kim⁴, Tae-Woo Lee⁴,
Valeria Lauter^{2*}, Bin Hu^{1*}

¹ Department of Materials Science and Engineering, University of Tennessee, Knoxville, TN 37996, USA

² Neutron Scattering Division, Neutron Sciences Directorate, Oak Ridge National Laboratory, Oak Ridge, 37831, Tennessee, USA

³ Center for Nanophase Materials Sciences, Oak Ridge, TN, 37831, Tennessee, USA; Neutron Scattering Division, Oak Ridge, TN, 37831, Tennessee, USA

⁴ Department of Materials Science and Engineering, Seoul National University, Seoul, 08826, Republic of Korea

* Corresponding author.

E-mail address:

lauterv@ornl.gov

bhu@utk.edu

Responses to Reviewer #1

Review comment

The manuscript titled “Spin Switchable Optical Phenomena in Rashba Band Structures through Intersystem Crossing in Momentum Space in Solution-Processing 2D-Superlattice Perovskite Film” by Bogdan Dryzhakov et al. reported the spin switchable phenomena in the multiferroic perovskite/Co interface, where the circularly polarized luminescence (σ^+ and σ^-) could be switched via changing the magnetic field direction (+B and -B). The momentum symmetry breaking of the Rashba band structure in this system with long orbital polarization lifetime and long-range spin-orbit interaction, provides a possible approach for developing potential spin devices. However, the quality of this manuscript does not meet the high standards of Nature communication. The followings are the main issues about this manuscript.

Author response

We thank the reviewer for the effort in reviewing our manuscript and providing constructive comments. We have addressed all review comments and provided the point-by-point responses, and also revised our manuscript accordingly.

Review comment 1

I have doubts about the data in this manuscript. Specifically, what is the DOP of Figure 3c? The ones in both the text and Figure 3d are 3.1%, while that in Figure 3c is 3.3%. Figure 5 also has this problem, and the values stated in the main text and the figures are different.

Author response

We thank the reviewer for the careful reading and for pointing out our typos caused by our careless typing. We have corrected all typos in the revised manuscript. Here are the specific corrections.

In Figure 3c, the DOP was mistakenly labeled as 3.3% while it should be 3.1% as it was indicated in both Figure 3d and the text. Similarly, for Figure 5, there was a typo. The figure displayed a value of 0.8%, while the text is labeled as 1%. This oversight was unintentional, and we have since amended the text and figure to accurately represent the value as 1%.

To provide more clarity, the left panels of Figure 6a,b were derived from Figure 2d (DOP=0.8%) and were meant to serve as a comparative purpose. Although Figures 2 and 6 represent measurements from separate films taken at different occasions, the thin films were prepared with identical procedures and especially were shown very similar phenomena. The noted confusion in Figure 6 stemmed from incorporating the data from Figure 2d into Figure 6 as an adjacent comparison. However, we also had an identical measurement for the film presented in Figure 6 that represents the 1% DOP written in the text. In our revised version, we have replaced the data in Figure 6 with the accurate measurement corresponding to that film, which aligns

with the DOP value of 1%. We appreciate the keen observation of the reviewer and have made the results more coherent by using the data from the same film in Figure 6 in this revised manuscript.

Review comment 2

Figure 3a does not clearly express the experimental design process and there are redundant lines.

Author response

We appreciate the feedback on Figure 3a regarding its presentation. Acknowledging the shortcomings of the original figure, we have taken steps to enhance its depiction. Firstly, we have revised Figure 3a to offer a more detailed schematic on the operation of measuring DOP with less visual clutter. Additionally, the methods section “Circularly Polarized Photoluminescence” now describes the experimental design of this measurement, indicating the experimental setup, the polarizers and optics used, and the operation process. These revisions provide a clearer representation of the experimental design process.

Review comment 3

The testing temperature for other data in the article is 5K, while Figure 4 shows 10K, which may have some differences. For the completeness and reliability of the data, it is recommended to retest at a temperature of 5K.

Author response

We thank the reviewer for this comment. We would like to explain why 5K and 10K were used for spectral measurements and polarized neutron reflectometry measurements. 5K is the lowest temperature of our helium closed cycle cryostat within our optics laboratory. 10K is the lowest temperature from the cryostat system integrated neutron system to allow laser beam excitation at Oak Ridge National Laboratory. The reason that we were using the lowest temperatures (5K and 10K) at our optics lab and neutron station (Oak Ridge National Lab) was to maximize the experimental reliabilities. In addition, our other optical studies have shown that 5K and 10K demonstrate very similar circularly polarized luminescence in such crystalline 2D perovskites. Also, we have added additional details to highlight the uniqueness of our measurement to the supplemental information document as Figure S4 and also a schematic diagram in Figure 5 in this revised manuscript. For the convenience to the reviewer, here we included the detailed measurement setup of polarized neutron reflectometry (Figure 4S).

Figure S4: Polarized neutron reflectometry with *in-situ* polarized photoexcitation experimental setup. The circularly polarized 405 nm continuous-wave laser is expanded onto the thin film sample, which sits inside a cryostat with sapphire optical window. Cobalt layer spin is aligned with an in-plane external magnet field. A polarized neutron beam is reflected carrying both magnetic and nuclear scattering information corresponding to depth-dependent magnetization.

Review comment 4

A small question is raised here. We have previously tested a 10nm Au layer, but it seems that it is not semi-transparent. Are 15nm Au and 17nm Co sure to be semi-transparent? Is there optical photo that can be added to the supplement information to prove?

Author response

Thanks to the reviewer for this question. Yes, the combination of 15nm Au and 17nm Co is indeed semitransparent. We have added the optical photo under white light condition to visually represent the appearance of our film the supplemental information in Figure S1 in this revised manuscript. Also, we have measured the optical transmission of our sample, added to the Supplemental Information as Figure S1 in this revised manuscript. As shown in Figure S1 also attached below, it is clear that the combined 15nm Au and 17nm Co allows the transmission of laser beam excitation of 405 nm and 343 nm, and also the transmission of photoluminescence peaked at 540nm.

Figure S1: (a) Optical photo under white light of transparent semiconductor/metal thin films on glass substrate (Au/Co/2D-perovskite/Glass). (b) Transmission curve of the thin film. The measurement uses a white light Xenon lamp, whose light is filtered into a narrow band using gratings. The power transmitted at each wavelength, with respect to a blank, is measured using a power meter and calculated as $T(\%) = (P_{\text{blank}} - P_{\text{film}}) / P_{\text{blank}}$. Glass absorption is significant around 300 nm, and the hybrid perovskite sharply absorbs at the 525 nm band-edge. The combined 17 nm Co and 12 nm Au layers are semi-transparent, allowing for 20% to 30% light transmission around the photoluminescence (PL) peak.

Review comment 5

Line 194, 20 degrees instead of 20 centigrade.

Author response

Thanks to the reviewer for this careful review. Now, we have corrected this error: replaced "20 centigrade" with "20 degrees" (line 21 on page 13, in methods).

Review comment 6

Line 267, Figure 4d is not visible in the figure, it should be Figure 5d.

Author response

Thanks to the reviewer again for such careful review. The incorrectly referenced figure number has been corrected in this revised manuscript.

Review comment 7

Finally, the main problem of this work is that the degree of polarization is so low that can be considered within the error range, thereby repetitive experiments are strongly necessary to verify the existence of such polarization.

Author response

Thanks to the reviewer for this comment. Here, the circularly polarized luminescence are purely generated by the helical ordering effects of light-emitting excitons populated in Rashba band structures. In such case, the degree of circular polarization is not low. Furthermore, we would like to emphasize that our magnetic field effects of CPL provide two important points. First, our magnetic field effects of CPL confirm that the observed CPL are indeed generated by the helical ordering effects of light-emitting excitons. We should note that it is very often that circularly polarized luminescence come from birefringence in crystalline materials where anisotropy structures are formed, which is an artifact. If circularly polarized luminescence come from birefringence, circularly polarized luminescence cannot show magnetic field effects. Clearly, our magnetic field effects provide a unique method to confirm whether circularly polarized luminescence come from light-emitting exciton carrying circular orbital momentum or from birefringence. Second, our magnetic field effects of CPL provide the first direct evidence that the helical order of light-emitting excitons within Rashba band structures is spin-dependent, which provides the necessary condition to demonstrate spin-switchable helical order of light-emitting excitons based on 2D-superlattice perovskite films.

Furthermore, we employed Transient Absorption Spectroscopy (TAS) with pumping laser wavelength of the same wavelength and polarization conditions as in photoluminescence measurements. The TAS results validate the subtle spin-switchable circularly polarized luminescence observed, where a spin-switching phenomena can be seen upon application of magnetic field. Within the context of this study, we credit the rapid process of spin switching to strong Rashba-type interactions. The revised manuscript includes these new results (Figures 4, S2, and S3) and discussions from TAS experiments, detailed with a focus on the spin-switching and stable spin-polarization decay. The added new results enhance the reliability and robustness of our findings, and we believe they adequately address the concerns raised.

Responses to Reviewer #2

Review comment

This work describes a study of magnetic field control of circularly polarized photoluminescence in two-dimensional hybrid perovskites/ferromagnetic heterostructures. Beyond the ordinary circularly polarized photoluminescence that has been widely reported in hybrid perovskite thin films upon circularly polarized light illumination (Fig. 2), the highlighted work stated by the authors is that when the perovskite thin film is interfaced with the ferromagnetic, Co layer, the circular polarization of photoluminescence can be altered upon the direction of an applied external magnetic field. The reversal of circular polarization has been attributed to the intersystem crossing between spin-polarized sub-bands caused by the Rashba splitting in hybrid perovskite films. Such a switchable behavior can also persist up to room temperature which is also concluded by the presence of super long-range spin-orbit interactions. However, the main conclusion drawn from this manuscript is not validated since this phenomenon can be simply described by a general magneto-optical Kerr effect that has been well accepted elsewhere. The observed phenomenon has nothing to do with the proposed Rashba state (no direct evidence for this statement either) and the spin-flip mechanism between spin sub-bands. Since this is the key statement in this manuscript, I cannot recommend the publication of this work in Nature Communications.

Author response

We thank Reviewer #2 for the comments on our work on the magnetic field control of circularly polarized photoluminescence in 2D-superlattice perovskite film interfaced with a ferromagnetic Co layer. We would like to emphasize that our studies are not related to magneto-optical Kerr effect at all, based on two important experimental points. First, the circularly polarized luminescence signals that we measured are generated by our 2D-superlattice perovskite film (not the excitation beam reflection from the magnetic Co substrate). Second, when the magnetic Co substrate is not used, a magnetic field can still change the degree of circular polarization of circularly polarized luminescence in the 2D-superlattice perovskite film. Third, our early studies of spin-polarized neutron scattering have found that circularly polarized photoexcitation-induced orbitals in hybrid perovskite films can directly interact with the spins in the magnetic Co substrate, generating optically induced magnetization in hybrid perovskite film (Wang, Miaosheng, et al. *Advanced Science*. 8.11 (2021): 2004488.).¹

Review comment 1

While the reviewer appreciates that a superlattice-like high-quality 2D hybrid perovskite thin film is demonstrated, the authors made an incorrect statement that the observation of circularly polarized photoluminescence implies the presence of the Rashba effect in their thin films. It is

not always valid. A strong spin-orbit coupling in hybrid perovskite will solely generate a J state in the CB that can be responsible for the selective circularly polarized light absorption and circularly polarized photoluminescence, similar to that in the prototypical GaAs system. No Rashba state is necessarily needed to enable this phenomenon. Please see Odenthal et al., Nat. Phys. 13, 894–899 (2017). A cross-check that can unambiguously prove the Rashba effect needs to be presented, such as indirect band transition probed by ultrafast PA spectroscopy or circular photogalvanic effect, etc.

Author response

We thank the reviewer for the time in reviewing our manuscript and also the comments on Rashba band structures. Here, we would like to mention several points about the Rashba band structures for hybrid perovskites generally and for the 2D-superlattice perovskite film under investigation: (1) The hybrid perovskites such as 2D-superlattice perovskite film possess strong spin-orbital coupling and symmetry breaking, providing the necessary condition to establish Rashba band structures.² (2) Rashba band structures have been experimentally^{3,4,5} supported by circular photogalvanic effect in hybrid perovskites and (3) theoretical models^{6,7} have been predicted as well for the hybrid perovskites. (4) When symmetry breaking is absent, J states cannot generate a circularly polarized luminescence because of lacking degeneracy between left-handed and right-handed J states. When symmetry breaking is present, J states can then generate a circularly polarized luminescence because the degeneracy between left-handed and right-handed J states is lifted. Therefore, the observed circularly polarized luminescence presents an experimental indication to support Rashba band structures in 2D-superlattice perovskite film. Indeed, GaAs does possess Rashba band structures.⁸ (5) Our magnetic field effects confirm that the observed circularly polarized luminescence represent Rashba band structures in 2D-superlattice perovskite film. (6) Our added new result from pump-probe transient absorption elucidates that right/left-handed circularly polarized photoexcitation causes different circularly polarized photoinduced absorption in 2D-superlattice perovskite film (Figure 4 in the revised manuscript). This provides further evidence to support Rashba band structures.

Review comment 2

The field control of circular polarization is mistakenly interrupted. The observed signals would be simply generated by the magneto-optical Kerr effect (MOKE) from the cobalt at the Co/perovskite interface. There are two components in the MOKE, one real part is about the rotation of linear polarization, i.e., Kerr rotation, and one imaginary part is about a change of the ellipticity, i.e., Kerr ellipticity. Please see Qiu & Bader, Rev. Sci. Instrum. 71, 1243–1255 (2000). The amplitude of Kerr ellipticity changes as a function of wavelength, following the magnetic hysteresis loop of the ferromagnetic layer. In this work authors only reported the results at two opposite magnetic fields. You may imagine that if a full loop of circular polarization is measured, it will follow exactly the magnetic hysteresis of the Cobalt layer.

A similar concept has been also widely employed in the field of 2D magnets to characterize their magnetic properties using reflectance magneto-circular dichroism (RMCD) microscopy, see Huang et al., Nature Nano. 13, 544–548 (2018).

Please note that the authors used a typical longitudinal MOKE/RMCD technique by which the PL is collected at the incident angle of 20 degrees to the normal plane. If the incident angle is bigger, a larger MOKE/RMCD signal would be obtained. There is irrelevant to the proposed Rashba effect in hybrid perovskite thin films that the authors stated.

Author response

Thank the reviewer again for comments. As we explained in the above response, we can confirm that Rashba band structures are responsible for our observed phenomena: spin-switchable circularly polarized luminescence circularly polarized in 2D-superlattice perovskite film. We should note that our measurements monitored the circularly polarized luminescence (peaked at about 530 nm), not the reflection of incident circularly polarized laser beam at 343 nm. Here, we like to mention an additional experimental point to further confirm the effects from Rashba band structures (added as Figure S5 in the revised manuscript). Linearly polarized laser excitation is referenced as a control condition to highlight the selective nature of the polarized neutron reflectometry (Figure 5) and circular polarized emission (Figure S5). The added results using linearly polarization excitation exhibits minimal optical and magnetic interaction with the magnetized interfacial Cobalt layer. Also, our circularly polarized pump-probe transient absorption indicates that there is a prompt spin switching to clearly support the effects of Rashba band structures discussed in the manuscript [sub-section “Dynamic monitoring of the spin polarization” with accompanying results in Figures 4, S2, and S3].

Finally, our polarized neutron reflectometry (PNR) results offer essential insight into the proposed mechanism. PNR demonstrates optically induced magnetization within the perovskite layer when exposed to circularly polarized photoexcitation. The observed magnetization of the 2D-perovskite's photoexcited orbitals is selectively influenced by the circular polarization of the light. This photoinduced magnetization lends significant weight to the interpretation of Spin-Rashba effects by demonstrating a long-range spin-orbit interaction. The interacting and directional coupling with cobalt spin is able to play a significant influence on spin-polarized light emission. With these observations we emphasize switchable spin-polarized band structures.

Review comment 3

Temperature-dependent data can be also explained by the MOKE response due to the reduced magnetization of the Co layer at 300K. By conducting a full magnetic hysteresis of the loop, the authors should observe a narrower loop because of the reduced coercive field in the Co film.

Author response

We performed multiple polarized neutron reflectometry (PNR) experiments on several samples at 300K, 14K and 10K. (PNR measures the absolute magnetization and its depth dependence) The magnetization of the Co layer did not show temperature variation or a “reduced magnetization at higher temperatures” such as 300K (See added figure below for PNR spin-asymmetry (SA) measured at 14K and 300K). We would like to remind here that PNR is magnetization vector magnetometry with depth resolution of 0.5 nm. There was no difference between the two measurements, thus confirming that there is no reduction in M of Co layer at 300 K.

The figure above compares the PNR spin asymmetry measured at 14K and 300K in our recent polarized neutron reflectometry experiment, to discuss the impact of temperature differences on magnetization.

Review comment 4

Time-dependent PL results in Fig. 2 don't support the conclusion for the intersystem crossing at the Rashba effect. The authors need to conduct the time-dependent PL measurement also in their heterostructure devices. If there is any intersystem crossing or spin flip between spin-polarized subbands that dominate this process, the circular polarization of PL emission would change as a function of time at opposite field directions.

Author response

We appreciate the reviewer's insights regarding the time-dependent PL results. It's essential to clarify a few points to address this comment accurately.

Time-resolved CPL predominantly tracks the evolution of spin states through emitted luminescence but may also entangle contributions from intersystem crossing. To probe the intersystem crossing dynamics more directly between Rashba-split bands, we employed Transient Absorption Spectroscopy (TAS) (Figure 4 added into the revised manuscript). Strong Rashba-type interactions are typically dominant in the early-time dynamics post-excitation, which can benefit from the picosecond resolution of TAS while gaining direct insight into the spin polarization dynamics and intersystem crossing.

The added TAS results confirm the spin polarization in the 2D-perovskite due to magnetization in the cobalt, with prompt spin flipping. The direction of spin polarization correlates with the magnetization direction of the cobalt, providing further evidence for the intersystem crossing mechanism governing these dynamics.

Review comment 5

The authors need to perform more control experiments to underpin the interfacial effect between the Co and hybrid perovskite thin films. One control experiment is to insert a thin Cu spacer layer (~2-5nm) between Co and hybrid perovskite to identify whether there is an emergent pseudo magnetic moment by magnetic proximity effect, transferring of the orbital angular momentum, or direct exchange coupling.

Author response

Thanks the reviewer for the effort/time in reviewing our manuscript. In response to the suggestion “to perform more control experiments” we would like to draw reviewer’s attention to our foundational findings in our previous report using polarized neutron reflectometry and circularly polarized laser excitation to demonstrate and analyze the interfacial effects between spin states in the hybrid perovskite thin film layer that interact with the ferromagnet Co.¹ That work followed our earlier studies which observed a prominent magnetodielectric coupling effect in the perovskite/Co interfacial layer, attributable to interactions between the spins on the ferromagnetic Co surface and orbitals on a hybrid perovskite surface layer.^{9,10}

The optically induced magnetization found in this study using the *depth-dependent* polarized neutron reflectivity *directly* demonstrated a uniform magnetization across the *entire* thickness of the perovskite film, consistent with a long-range spin-orbit exchange interaction and not a near-field decay from proximity effect (Figure 5 in the revised manuscript). This conclusion is further strengthened by the requirement of circularly polarized excitation to induce this magnetization, noting the conditionality to photoexcite circularly polarized orbitals in the hybrid perovskite layer. Lastly, our added TAS data utilizes circularly polarized pump and probe to target spin-polarized carriers to demonstrate the effects of Rashba band structures (Figure 4 in the revised manuscript).

Responses to Reviewer #3

Review comment

The work proposed about spin switchable circularly luminescence is relevant to the spin optoelectronics research community and goes in the current direction of modulating this optical property in novel materials. However, to fully support the conclusions and claims, a major revision of the manuscript with additional evidence/experiments is needed before considering for publication. There are missing some important details about the methodology and control experiments. Here the critical points that should be addressed:

Author response

We sincerely appreciate Reviewer #3 recognition of the novelty of our work in the context of spin optoelectronics, and also the feedback that helps enhance the rigor and depth of our manuscript. In response to this feedback, and in line with other reviewers' comments, we have clarified our methodology and expanded on critical aspects of our approach for reproducibility. Additionally, we have included transient absorption experiments to further elucidate the spin-switchable phenomena from Rashba band structures in 2D-superlattice perovskite film.

Reviewer #3 comment 1

In the methods section, authors mentioned that “on top of the 4,4-DFPD2PbI4 film prepared on glass substrate, semi-transparent layers of cobalt (17 nm) and gold (12 nm) are thermally evaporated.” Then, some of the sentences along the text are imprecise: Abstract - “film spin-coated on ferromagnetic cobalt (Co) substrate”; introduction - “respectively, for a 2D-superlattice perovskite film prepared on a ferromagnetic Co surface.”, and this is a key point since it explains how the system under study is made. Following this line, authors should provide experimental data proving that the Co/Au thermal deposition is not damaging the perovskite layer.

Author response

Thank the reviewer for the detailed comments. The film preparation of spin-coating, vacuum, and annealing, and a deposition of Co and Au layers, follows a well-developed and established protocol based on fabrication of multiple samples, their characterization with X-rays, AFM and PNR resulting in the detailed optimization of various conditions for the controllable sample preparation. This has become a standard process for our group, which was originally developed several years ago to optimize films for PNR measurements. Ferromagnetic heterostructures made from 17 nm thick cobalt have also been optimized for this purpose, with 12 nm thick gold used as a balance between effective capping layer and maintaining transparency.

As it is explained in the text (see Methods) PNR is a very powerful depth-sensitive method:

“PNR is a highly penetrating depth-sensitive technique to probe the chemical and magnetic depth profiles with a resolution of 0.5 nm. The depth profiles of the nuclear scattering length density (NSLD) and magnetic scattering length density (MSLD) precisely correspond to the depth profile of the chemical and IP magnetization vector distributions on the atomic scale, respectively^{11,12,13}. Based on these neutron scattering merits, PNR was applied simultaneously and nondestructively to characterize chemical and magnetic nature of buried interfaces.¹⁴

We added a text in page 15:

“The thickness of each individual constituent layer, interfacial roughness, and magnetization depth profile were obtained in detail for all samples (See figure 5). The SLD of the perovskite film (which measured chemical composition and density of the film) shows uniform distribution confirming the excellent quality of the film. The roughness at the perovskite/ Co and Co/Au interfaces is 3.5nm and 2.5 nm respectively.”

Review comment 2

In Figure 1e, the PL spectrum at 5K shows multi-peaks on the right and left sides of the main emission peak. Authors commented about the presence of these multi-peaks, but not about their origin. How do authors explain the appearance of these peaks? Usually in high quality perovskite thin films the photoluminescence (PL) spectrum at low temperatures is simply narrower than at room temperature, but it is not common to observe new features in both sides of the photoluminescence emission. Even more, the shape of the PL spectrum at 5K in Figure 1 is completely different than the one in Figure 2 a-b. The same happens for the room-temperature PL spectrum but less marked. Authors should explain the reasons why this is happening in more detail.

Author response

We thank the reviewer for this detailed comment. We have also noticed in our studies that the PL spectrum can show clear shoulders at low temperature. We are currently investigating this phenomenon as a further study towards further papers. In general, at low temperature the spectrum could be narrow, and the shoulders become clearer. At room temperature the spectrum could be broader, and the shoulders become disappear. Regarding σ^- PL spectrum has clearer shoulders as compared to σ^+ PL spectrum at 5 K, this means that the spin-up and spin-down band structures have different phonon coupling, leading to different spin-phonon coupling in 2D-superlattice perovskite film. We are currently investigating this hypothesis by using magnetic field effects of circularly polarized pump-probe spectroscopy. We thank the reviewer for this comment again.

Review comment 3

In figure 1g, it is shown a “PL change in %”, but in the text authors refer to “anisotropic PL intensities”. Therefore, it is misleading. Mainly because there is no methods section explaining how the PL measurements were carried out: polarization only in the excitation path?

Spectrometer/detector used? In the data analysis did authors use absolute intensity or integrated area?

Author response

Thank the reviewer for the questions. A methods sub-section titled “Polarized Light Emission” has been added to explain the methodology used in light emission measurements. Also, the text describing Figure 1g has been adjusted to describe the measurement conditions more clearly.

Review comment 4

Regarding Figure 3, authors comment “before applying the ferromagnetic Co layer [...] CPL-DOP is 0.3%”, but there is not data showing this as well as the CPL after Co/Au deposition at 0T magnetic field. Indeed, looking at Figure 2, this value is smaller than the 0.6/0.8% reported there. Therefore, is it a question of reproducibility among different samples? But most importantly, why did the authors carry out the experiments in Figure 3 and 5 only at room temperature and not at 5K, where, as shown in Figure 2a-b, the DOP values were already higher and therefore, more accurate results could be obtained?

Author response

Thank you to the reviewer again for the detailed comments/questions. We noticed there are small variations in the CPL-DOP values between different samples. However, the circularly polarized luminescence and magnetic field effects are true phenomena and reproducible phenomena. Regarding why we did magnetic field effects at room temperature, our primary motivation was to focus on the room temperature phenomena, which provides more practical implications to optoelectronic applications.

Review comment 5

Comparing Figures 3 and 5 and most concretely the PL spectra, the shape of the PL spectra shown in panels Figure 3b-c and Figure 5a-b-left and different than the ones in Figure 5a-ac-right. It seems that there is not the same background signal, since intensity counts at 450 nm are higher than at 650 nm in the first ones and in the second both are equal

Author response

We thank the reviewer for the comments/questions. The weak background signal is caused by the glass encapsulation layer for our samples. Specifically, for room temperature measurements, the Co/Au/perovskite samples were encapsulated by glass plates which give a small background signal. During low temperature (5K) measurements, our samples were not encapsulated. In light of this, we also noticed the inconsistency in Figure 5. The left panel plot of Figures 5a and 5b is taken from Figure 2d for incorporated for comparative analysis. In the right panel of Figures 5a and 5b, the background signal shoulder is less emphasized due to the absence of glass encapsulation, while the left panel taken from Figure 2d was glass encapsulated. Incidentally, we also had an identical measurement for the film presented in Figure 5. In our revised version, we have replaced the comparative panel plots (left panels in Figure 5a and 5b) with the accurate measurement corresponding to that film. We have inserted a comment in the experimental section to note the origin: [*The observed shoulder on the high-energy side of the*

PL spectra in certain figures originates from glass, especially pronounced when the PL signal is weaker due to conditions such as the presence of Co/Au layers or upon glass encapsulation].

Review comment 6

There is not enough detail provided in the methods for the work to be reproduced. Paragraphs regarding absorbance and PL measurements, lifetime and SHG are missing. For example, in Figure 3a, the scheme indicates a 20° angle for incident excitation and collected signal, but there is no explanation. This Reviewer supposes that it should be related to the experimental set-up, but at the same time, the incident/collection angle should affect the PL spectrum since these perovskite thin films could suffer from reabsorption effects.

Author response

We thank the reviewer for this comment. Yes, the 20° angle was due to the limitation of our setup. However, we have manually examined that changing angle does not lead to noticeable reabsorption. In addition, the descriptions for absorbance, photoluminescence, lifetime, and SHG have been added to the methods sections in this revised version.

Review comment 7

One key point in research is the reproducibility of the results. Along the text, this Reviewer is not able to know in how many samples did the authors observe the “Spin Switchable Optical Phenomena”, considering the fabrication procedure: spin coating/vacuum/annealing steps for the perovskite film preparation and thermal evaporation for “17 nm” Co and “12nm” Au deposition.

Author response

Thank the reviewer for the comments. The film preparation of spin-coating, vacuum, and annealing, and a deposition of Co and Au layers, follows a well-developed and established protocol based on fabrication of multiple samples, their characterization with X-rays, AFM and PNR resulting in the detailed optimization of various conditions for the controllable sample preparation. This has become a standard process for our group, which was originally developed several years ago to optimize films for PNR measurements. Ferromagnetic heterostructures made from 17 nm thick cobalt have also been optimized for this purpose, with 12 nm thick gold used as a balance between effective capping layer and maintaining transparency.

Review comment 8

Minor issues that should be addressed:

-Please correct misspellings and wrong notations along the text, e.g. “2D-sperlattice”, “20°C”; and the format/content of some of the references.

Author response

We thank the Reviewer for pointing out the oversight and have amended the manuscript, fixing these misspellings and the reference section.

References

-
- ¹ Wang, M. *et al.* Optically Induced Static Magnetization in Metal Halide Perovskite for Spin-Related Optoelectronics. *Advanced Science* **8**, 2004488 (2021). <https://doi.org/10.1002/advs.202004488>
- ² Even, J., Pedesseau, L., Dupertuis, M. A., Jancu, J. M. & Katan, C. Electronic model for self-assembled hybrid organic/perovskite semiconductors: Reverse band edge electronic states ordering and spin-orbit coupling. *Physical Review B* **86**, 205301 (2012). <https://doi.org/10.1103/PhysRevB.86.205301>
- ³ Becker, M. A. *et al.* Bright triplet excitons in caesium lead halide perovskites. *Nature* **553**, 189-193 (2018). <https://doi.org/10.1038/nature25147>
- ⁴ Zhai, Y. *et al.* Giant Rashba splitting in 2D organic-inorganic halide perovskites measured by transient spectroscopies. *Science Advances* **3**, e1700704 (2017). <https://doi.org/10.1126/sciadv.1700704>
- ⁵ Zhang, L. *et al.* Room-temperature electrically switchable spin-valley coupling in a van der Waals ferroelectric halide perovskite with persistent spin helix. *Nature Photonics* **16**, 529-537 (2022).
- ⁶ Kepenekian, M. *et al.* Rashba and Dresselhaus Effects in Hybrid Organic-Inorganic Perovskites: From Basics to Devices. *ACS Nano* **9**, 11557-11567 (2015). <https://doi.org/10.1021/acsnano.5b04409>
- ⁷ Kim, M., Im, J., Freeman, A. J., Ihm, J. & Jin, H. Switchable $S = 1/2$ and $J = 1/2$ Rashba bands in ferroelectric halide perovskites. *Proceedings of the National Academy of Sciences* **111**, 6900-6904 (2014).
- ⁸ Koralek, J., Weber, C., Orenstein, J. *et al.* Emergence of the persistent spin helix in semiconductor quantum wells. *Nature* **458**, 610-613 (2009). <https://doi.org/10.1038/nature07871>
- ⁹ Li, M. *et al.* Magnetodielectric Response from Spin-Orbital Interaction Occurring at Interface of Ferromagnetic Co and Organometal Halide Perovskite Layers via Rashba Effect. *Advanced Materials* **29**, 1603667 (2017). <https://doi.org/10.1002/adma.201603667>
- ¹⁰ Li, M. *et al.* Interaction Between Optically-Generated Charge-Transfer States and Magnetized Charge-Transfer States toward Magneto-Electric Coupling. *The Journal of Physical Chemistry Letters* **6**, 4319-4325 (2015). <https://doi.org/10.1021/acs.jpcclett.5b01838>
- ¹¹ Blundell, S. J. *et al.* Spin-orientation dependence in neutron reflection from a single magnetic film. *Physical Review B* **51**, 9395-9398 (1995). <https://doi.org/10.1103/PhysRevB.51.9395>
- ¹² Lauter-Pasyuk, V. *et al.* Magnetic off-specular neutron scattering from Fe/Cr multilayers. *Physica B: Condensed Matter* **283**, 194-198 (2000). [https://doi.org/10.1016/S0921-4526\(99\)01938-9](https://doi.org/10.1016/S0921-4526(99)01938-9)
- ¹³ Lauter-Pasyuk, V. Neutron grazing incidence techniques for nano-science. *Collection SFN* **7**, s221-s240 (2007).
- ¹⁴ Lauter, V., Lauter, H. J. C., Glavic, A. & Toperverg, B. P. Reflectivity, Off-Specular Scattering, and GISANS Neutrons in Reference Module in Materials Science and Materials Engineering p. 1-27 (Oxford: Elsevier, 2016).

Responses to Reviewer #1

Reviewer #1 (Remarks to the Author):

My questions and concerns have been addressed by the authors. The manuscript has been satisfactorily revised. Based on the major improvement made, I would like to pass it for acceptance.

Author Response

We thank the reviewer for their support. To summarize, we improved description of the measurement setup of polarized neutron reflectometry (Figure S4), measured optical transmittance (Figure S1), and explored the DOP effect with TAS measurements (Figures 4, S2, and S3). We appreciate this feedback that strengthened our study.

Responses to Reviewer #2

Reviewer #1 (Remarks to the Author):

The authors have made deliberate efforts to respond to all the reviewers' comments. Some of the added control experiments have partially resolved my concerns but not all, particularly for the key statement: spin-switchable magneto-CPL assisted by the proposed Rashba state. Therefore, I cannot recommend the publication of this work until my concerns are fully resolved. Below I list my unaddressed concerns following the sequence in the rebuttal letter.

Author Response

The reviewer's acknowledgement of our effort to respond and answer all initial comments is appreciated. We also appreciate each new comment and address the concerns in this response letter. Importantly, our study builds upon a substantial body of evidence regarding the Rashba effect in non-centrosymmetric hybrid perovskites, including a few of our foundational papers to photo-induced coupling effects in work reported here ^[1,2,3] and one recent publication demonstrating the Rashba effect on the specific material (4,4-DFPD₂PbI₄)^[4], same as our study here.

While we are certainly interested in these reviewer's suggested measurements, some of the additional experiments suggested are beyond the primary focus of our current work and would rather result in follow up publications. We argue that the central message of our work does not explicitly rely on this additional undertaking since it would unreasonably expand the scope of our paper. We value the expertise of the reviewer and provide our reasoning to the remaining concerns in the following responses.

Review comment

Comment 1: *There are indeed many literature reports about the presence of the Rashba state in layered hybrid perovskite. But this cannot stop me from asking for direct experimental evidences of the Rashba effect in the currently used 2D perovskite thin film. Other groups reported the Rashba effect in their 2D perovskite thin films with a different composition and symmetry breaking, not the current one. So please provide:*

(1) DFT calculations about the spin-splitting state in the 4,4-DFPD₂PbI₄ film and describe the Rashba parameter in the unit of eV/Å.

Author Response

While our current study focuses primarily on experimental results, we find that the requested computational support material of the same material subject (4,4-DFPD₂PbI₄) is presented in a recent Nature Photonics article (2022) that conducts the DFT calculations and report a high Rashba parameter (E_R) of 62meV.^[4] This theoretical prediction emphasizes and infers that room-temperature operation is feasible. Additionally, our study explores the Rashba-split states through multifaceted optical experiments, thus providing the requested various experimental evidence of the Rashba effect in the currently used 2D perovskite thin film. Second harmonic generation demonstrates the necessary symmetry breaking for band splitting, and circularly polarized luminescence serves as a key indicator of spin-momentum locking in the split bands. Moreover, selectivity of PNR-measured magnetization under circularly polarized photoexcitation and no

magnetization under linearly polarized photoexcitation is our evidence of Rashba-type spin-orbit coupling.

Review comment

(2) Experimental proof of the spin-momentum relation, e.g., circular photogalvanic effect (CPGE) in the 4,4-DFPD₂PbI₄/Co heterostructure. The circular polarization-dependent pump-probe spectroscopy is inadequate to prove the presence of the Rashba state. I didn't see any feature of the indirect band transition. Please show the influence of the applied magnetic field on the spin-splitting sub-bands (energy shift, density state, etc.) via CPGE, leading to the field-dependent CPL polarization.

Author Response

The presence of the Rashba state in 4,4-DFPD₂PbI₄ has already been demonstrated in [4]. The methods of probing spin-momentum coupling measurement with Circular Photogalvanic Effect (CPGE) and Linear Photogalvanic Effect (LPGE) were primary results of the 2022 Nature Photonics article, which studied the same material (4,4-DFPD₂PbI₄) and experimentally demonstrated spin texture switching and spin-momentum coupling in polarized excited electrons. To address the reviewer's comment and to better contextualize our work within the existing literature, we supplement our discussion in the introduction by adding the following statement:

“Rashba splitting in 4,4-DFPD₂PbI₄ was recently reported using circular photogalvanic effect to monitor spin-momentum coupling and spin texture switching, an observation which opens the possibility of spin operability through the presence of Rashba effects.” (page 3, lines 20-23)

In our current work, we focus on measuring optical analogs of spin-polarized states. Here, magnetic field effects of circularly-polarized pump-probe transient data provide the further experimental evidence to support Rashba band structures. Our early studies found that circularly-polarized photoexcitation can induce a magnetization in 3D perovskite film when interfaced with ferromagnetic Co surface, leading to an optically induced magnetization [Advanced Science, DOI: 10.1002/advs.202004488].^[1] We have observed a similar and enhanced phenomenon in our 2D-superlattice perovskite film (Figure 5 in this manuscript). Essentially, the polarization-selective optically induced magnetization reveals the interaction between the circularly-polarized orbital momentum in 2D-superlattice film and the spin dipoles on the Co surface. When considering the experimental geometry with the incident angle of 20° of photoexcitation beam illustrated in Figure C-1 below), the positive and negative magnetic fields (+B and -B) lead to smaller and larger optically induced magnetizations (M_1 and M_2) by interacting with circularly-polarized orbitals, respectively.

Figure C-1. σ is the magnetic dipole from circularly-polarized orbital. M_1 and M_2 are optically induced magnetizations at $+B$ and $-B$, respectively.

The optically-induced magnetizations cause a spin scattering for circularly-polarized excitons and consequently generate an intersystem crossing between spin-up and spin-down bands in Rashba band structures. At $+B$, the smaller optically-induced magnetization gives rise to a negligible intersystem crossing due to weaker optically-induced magnetization. In this situation, exciting spin-up and spin-down band structures by using σ^+ and σ^- excitations can lead to similar population dynamics in both spin-up and spin-down band structures due to negligible intersystem crossing, as experimentally shown in Figure 4 for magnetic field effects of circularly-polarized pump-probe transient absorption data. At $-B$, the larger optically-induced magnetization gives rise to an appreciable intersystem crossing due to stronger optically-induced magnetization. In this situation, exciting spin-up and spin-down band structures by using σ^+ and σ^- excitations can give rise to different population dynamics in spin-up and spin-down band structures because optically-induced magnetization can more and less effectively interact with spin-up and spin-down band structures to generate more and less intersystem crossing, as shown in Figure 4 for magnetic field effects of circularly-polarized pump-probe transient absorption data. Clearly, our magnetic field effects of circularly-polarized pump-probe transient absorption data further confirm that Rashba band structures are formed in our 2D-superlattice perovskite film with spin-dependent circularly-polarized optical phenomena. Finally, the transient absorption measurements with circularly polarized pump-probe beams show a handedness selective polarization with differences in their spin dynamics. When exciting with counter-polarized pump and probe beams, an applied magnetic field rapidly shifts the exciton to the other band and with a slower rise in degree of spin polarization compared to when using co-polarized beams.

The following revisions have been made to our manuscript:

“With an experimental geometry using an incident angle of 20° , the interaction between the circularly polarized light and the Co layer induces optically generated magnetizations whose magnitudes depend on the direction of the applied magnetic field (Figure S7).” (page 14, lines 2-5).

Figure S7: (a) σ is the magnetic dipole from circularly-polarized orbital. M_1 and M_2 are optically induced magnetizations at $+B$ and $-B$, respectively. The optically induced magnetizations, M_1 and M_2 , arise from this interaction and have magnitudes that depend on the direction of the applied magnetic field: smaller M_1 under a positive magnetic field ($+B$) and larger M_2 under a negative magnetic field ($-B$). (b) Rashba band structure with spin-up and spin-down bands and the intersystem crossing between these bands under different magnetic field directions. Under a negative magnetic field (-1 T), a larger optically induced magnetization enhances intersystem crossing between spin-up and spin-down bands in the Rashba band structure, leading to differences in population dynamics for σ^+ and σ^- excitations. Conversely, a positive magnetic field ($+1$ T) results in a smaller optically induced magnetization, causing negligible intersystem crossing and similar population dynamics for both spin polarizations. This variation in intersystem crossing with magnetic field direction provides further experimental evidence supporting the presence of Rashba band structures in our 2D-superlattice perovskite film. (c) Normalized photoluminescence spectra at different incident angles of excitation beam.

Review comment

Comment 3: Please just measure the M vs. T response in the 4,4-DFPD₂PbI₄/Co heterostructure using a SQUID magnetometer. The change of magnetization of the Co layer would be easily identified without involving the complicated PNR measurements.

Author Response

We sincerely appreciate the reviewer's suggestion to measure the magnetization of the 4,4-DFPD₂PbI₄/Co heterostructure using a SQUID magnetometer. Please note that magnetization measurements have already been provided and confirmed that there is no change of magnetization of the Co layer with temperature. Thus, additional SQUID measurements will not provide any additional information.

(1) Polarized Neutron Reflectometry is a precise and a depth-dependent method compared with SQUID, and here it is particularly crucial for heterostructure studies. By PNR, we provide concrete evidence there is no temperature dependence for cobalt. In figure 'a' below, we present the PNR spin asymmetry at both 14K and 300K of the 4,4-DFPD₂PbI₄/Co/Au thin film. A PNR spin-asymmetry (SA) is determined exclusively by the magnetization of the sample. Thus, if SA measured from the same sample at different temperatures, does not change, it means that there are no changes in magnetization. This is exactly as it would be measured with SQUID, VSM or other

method. Thus, the spin asymmetry provides precise information on the presence and orientation of magnetic moments within the layers of the heterostructure.

(2) Our foundational study (Wang, M., et. al., *Adv. Sci.* 2021, 8, 2004488), in which optically induced magnetism was first observed in a MAPbBr₃/Co/Au sample, we used SQUID to measure the magnetic hysteresis loop in the in-plane direction. The measurement from that paper is shown in figure b below. This gives confidence to the methods of materials preparation replicated in this manuscript (thermal evaporation of cobalt and spin coating of hybrid perovskite).

Figure C-2. Reference Magnetic measurements including (a) PNR spin-asymmetry (SA) of 4,4-DFPD₂PbI₄/Co/Au thin film and (b) SQUID measurement of previously reported sample adapted from Ref: Wang, M., et. al., *Adv. Sci.* 2021, 8, 2004488.

Review comment

Comment 4: The current data sets (only at +/- 1T) are inadequate to prove the authors' statement. Please systemically change the strength of the magnetic field (and near the coercive field of the Co layer) to see whether the proposed intersystem crossing truly correlates with the change of spin-state.

Author Response

We appreciate the reviewer's suggestion to systematically vary the magnetic field strength. However, we respectfully disagree that additional data points would provide meaningful insights in this particular case. The +/-1T measurements we've presented are in a saturation state of Co magnetization, which provides the clearest picture of how the perovskite layer responds to the magnetic field of the Co layer. At intermediate field strengths, including near the coercive field of the Co layer, the magnetic domain structure in Co layer develops into randomly oriented domains. This non-uniform field distribution would make it extremely challenging to draw clear conclusions about the correlation between intersystem crossing and overall spin-state changes. Our choice to focus on the +/-1T measurements is deliberate, as these conditions allow us to isolate the effect we're studying without the confounding influence of domain structures. We would also like to note that the amorphous cobalt thin films grown on a hybrid perovskite have quite small coercive field strengths.

The following revisions have been made to our manuscript:

"At ±1 T, magnetic saturation of the cobalt layer provides a uniform magnetic field that isolates the perovskite's intrinsic spin-state transitions by minimizing the influence of domain wall

movements and other complexities introduced by magnetic domain structures near the coercive field in the amorphous cobalt film.” (page 14, lines 16-19)

Review comment

Comment 5: If there is indeed a unique interfacial Rashba effect (not arise from the 4,4-DFPD2PbI4 itself) formed at the interface and modulated by the magnetic layer, did the authors conduct magneto-CPL measurement in a much thicker or thinner film? Does the polarization of field-dependent CPL remain the same? The interfacial effect should exist only within < 5-10nm near the interface.

Author Response

Our Polarized Neutron Reflectometry (PNR) measurements (Figure 5) reveal long-range interactions between spin and circularly polarized orbitals across the entire 70 nm film thickness. Polarized neutron reflectometry measures a depth profile of the film's thickness and shows that the optically induced magnetization runs consistently through the whole thickness of the film. This magnetization is what drives the magneto-CPL measurement results. Regarding film thickness variations, the results presented in this manuscript focus on a single material thickness, where the phenomena is depth resolved in PNR.

Responses to Reviewer #3

Review comment

Reviewer #3 (Remarks to the Author):

Authors have made some corrections and updates in the manuscript, but the response to some of the key questions from Reviewers is vague and not precise. Here the main points that should be addressed:

- Authors claim: the protocol to prepare the perovskite films “has become a standard process for our group” “we noticed there are small variations in the CPL-DOP values between different samples”. However, authors do not provide statistics and a concrete number of samples in which they have checked the reproducibility of the CPL-DOP values observed. Indeed, how can they state that the 0.8% values are not inside the error of the measurements?

Author Response

Our perovskite film preparation involves a multistep process, including A-site cation chemical preparation, crystal growth and cleaning, a four-step thin film processing, and deposition of Co/Au layers. While this protocol has become standard in our group, variations between samples can arise due to the complexity of the procedure. For instance, high-quality and high-purity single crystals of 4,4-DFPD₂PbI₄ are vital to preparing precursor solution for spin coating thin films with superlattice-like crystallinity; quality and purity are impacted by the age of single crystal precursors, size of single crystals, and adequate crystal rinsing. Indeed, the perovskite films with the highest superlattice-like crystallinity are reported in this manuscript as they are found to have the greatest DOP. Another challenge for sample-to-sample reproducibility is due to Cobalt's high melting point, which makes it challenging to precisely control the deposition rate, leading to slight deviations from the target thickness of 17 nm. This can impact the effective intensity of photoexcitation and the light collection efficiency.

Despite these sample-to-sample variations, the net change in CPL-DOP is observed consistently under applied magnetic fields for each sample, ruling out measurement error of changing excitation/collection polarizations. The observed values of 0.8% (Figure 2d) and 0.6% (Figure 2e) correspond to measurements with opposite handedness of circularly polarized light, a distinction that reflects the differential circular polarization dependence, and can therefore be treated as +0.8% and -0.6%. Importantly, the measured changes in intensity are consistent with the selected polarization states of light, indicating that the measurements are above the noise level and not within the experimental error. We validated the reproducibility of this contrast across multiple samples, which have differences in DOP magnitude but not in DOP handedness.

In Figure C-3, CPL signals of 5 different 2D-perovskite thin films with various dopant levels are shown to demonstrate the consistency of experimental measurement. Mixed Pb-Sn systems incur no net difference between σ^+ and σ^- light, meanwhile pure Pb or Sn materials do exhibit a net change, which is unmistakably above measurement error. Given the magnitude of measurement error is within the noise floor, the observed CPL-DOP values of approximately 0.8% are notable.

Figure C-3: CPL measurements of various 2D-perovskite thin films based upon a mixed Pb-Sn materials system. (Data from manuscript in preparation)

Review comment

- “The weak background signal is caused by the glass encapsulation layer for our samples.” Related to Figure 2. Then, authors should provide the signal from the glass as reference.

Author Response

Attached below is reference light emission spectrum of the amorphous glass substrate under 343nm laser excitation. The emission from glass appears as a high-energy shoulder of the room-temperature spectra in Figure 2d,e and Figure 3b,c. Figure C-4 has been added to supporting information as Figure S8.

The following revisions have been made to our manuscript:

The observed shoulder on the high-energy side of the PL spectra in certain figures originates from glass, especially pronounced when the PL signal is weaker due to conditions such as the presence of Co/Au layers or upon glass encapsulation (Figure S8). (page 14, lines 11-14)

Figure S8: Reference emission spectra of the amorphous glass substrate under 343 nm laser excitation is overlaid on the 4,4-DFPD₂PbI₄ PL. Emission from the glass appears as a high-energy shoulder in the room-temperature spectra shown in Figures 2d, 2e, 3b, and 3c.

Review comment

- Authors claim that the 17 nm Co/12nm Au layer is semitransparent but looking to Figure S1 at the excitation wavelength used for photoluminescence measurements, i.e. 343 nm, the transmittance is < 10%. It is really challenging to state that with this low transmittance the metallic film is semitransparent.

Author Response

To avoid any misunderstanding, we revise the manuscript to refer to the layer as 'partially transparent'. However, it is important to note that the <10% transmittance at 343 nm shown in Figure S1 is for the entire Au/Co/2D-perovskite/Glass stack, not just the Co/Au layers. This combined absorption involves the Co/Au layers, glass substrate, and the 2D-perovskite band-absorption. This validates the thin film system is sufficiently transparent to the 343nm laser, and that the Co/Au layers permit sufficient excitation power for photoexcitation in optical measurements. Language updated in main text: (page 7, line 13), (page 8, line 12), (page 14, line 1)

Review comment

- In the experimental section there are no details regarding the precursors amount (mg, mL, ...) used for the synthesis of the crystals. Authors should include these details.

Author Response

We appreciate this suggestion and have revised the experimental section to include the quantities of the materials used:

"Millimeter-sized crystal shards of 4,4-DFPD₂PbI₄ are grown by precipitation during slow cooling of precursor solution. Difluoropiperidine is added to solution of 10mL of hydroiodic acid (57%, unstabilized) with hypophosphoric acid (~1mL or until hydroiodic acid turns a pale yellow) at 0.1M, then stirred at 50°C for 30mins. Then lead iodide (99.999%) is added to the solution at a stoichiometric ratio (0.05M) and the solution is further stirred at 100°C until fully dissolved appearing as a pale-yellow solution. During cooling from 100°C to 60°C at a rate of 1°C/hr, the crystals begin to nucleate at ~85°C and are harvested at 60°C. The harvested crystals are washed and stirred in anhydrous methyl acetate three times then dried at 80°C under vacuum for 5hrs. (page 13, lines 3-10)

... These solution processing methods, crystal growth and thin film fabrication process, are illustrated in Figure S6." (page 13, lines 19-20)

A schematic of the single crystal growth and thin film fabrication process is provided (supporting information Figure S6). We would also like to note that this crystal growth procedure was detailed in our earlier work on the ferroelectric properties of this single crystal.⁵

Figure S6: Materials synthesis and preparation schematics. (a) Single crystal growth through temperature cooling, showing the resulting centimeter-sized, orange crystals. (b) Thin film fabrication process via spin-coating, including slow and fast spin steps, vacuum treatment, and thermal annealing, with the final vibrant colored thin film product shown.

Review comment

- *Authors claim:* “we have manually examined that changing angle does not lead to noticeable reabsorption” referring to the 20° angle used for the measurements, but there is no new graph, even just in the rebuttal letter showing this angle-independent behavior.

Author Response

In Figure C-4, we provide angle-dependent excitation data for the 4,4-DFPD₂PbI₄ thin film. This figure compares spectra collected at 20° and 70° angles of laser incidence, with emission collected at normal (0°). There is minimal variation in spectral profile between these angles, confirming the absence of significant reabsorption effects. Changing the angle primarily alters the probed volume within the thin film due to path length variations, which doesn't notably impact reabsorption for these measurements at the current film thicknesses. Given these results, we are confident that reabsorption does not significantly affect our findings. This figure has been added to supporting information as Figure S7c.

Figure C-4: Angle-dependent excitation spectra of 4,4-DFPD₂PbI₄ thin film with laser incidence at 20° and 70°, and emission collected at normal (0°).

-
- ¹ Wang M, et al. Optically Induced Static Magnetization in Metal Halide Perovskite for Spin-Related Optoelectronics. *Adv Sci* **8**, 2004488 (2021).
- ² Li, M. *et al.* Magnetodielectric Response from Spin–Orbital Interaction Occurring at Interface of Ferromagnetic Co and Organometal Halide Perovskite Layers via Rashba Effect. *Advanced Materials* **29**, 1603667 (2017). [https://doi.org:https://doi.org/10.1002/adma.201603667](https://doi.org/10.1002/adma.201603667)
- ³ Li, M. *et al.* Interaction Between Optically-Generated Charge-Transfer States and Magnetized Charge-Transfer States toward Magneto-Electric Coupling. *The Journal of Physical Chemistry Letters* **6**, 4319-4325 (2015). [https://doi.org:10.1021/acs.jpcllett.5b01838](https://doi.org/10.1021/acs.jpcllett.5b01838)
- ⁴ Zhang, L. *et al.* Room-temperature electrically switchable spin–valley coupling in a van der Waals ferroelectric halide perovskite with persistent spin helix. *Nature Photonics* **16**, 529-537 (2022).
- ⁵ Kim D, et al. Ferroelectric and charge transport properties in strain-engineered two-dimensional lead iodide perovskites. *Chem Mat* **33**, 4077-4088 (2021).